# Fragmentation of ice particles: laboratory experiments on graupel-graupel and graupel-snowflake collisions

Pierre Grzegorczyk[1,2], Sudha Yadav[1], Florian Zanger[1], Alexander Theis[3], Subir K. Mitra[1], Stephan Borrmann[1,3], and Miklós Szakáll[1]

[1]Institute for Atmospheric Physics, Johannes Gutenberg University, Mainz, Germany
[2]Laboratoire de Météorologie Physique (UMR6016)/UCA/CNRS, Aubière, France
[3]Particle Chemistry Department, Max Planck Institute for Chemistry, Mainz, Germany,

**Correspondence:** Miklós Szakáll (szakall@uni-mainz.de)

**Abstract.** Until now, the processes involved in secondary ice production which generate high concentrations of ice crystals in clouds have been poorly understood. However, collisions that involve rimed ice particles or crystal aggregates have the potential to effectively produce secondary ice from their fragmentation. Unfortunately, there have only been a few laboratory studies on ice-ice collision so far, resulting in an inaccurate representation of this process in microphysical schemes. To ad-
dress this issue, experiments were conducted at the Wind tunnel laboratory of the Johannes Gutenberg University, Mainz on graupel-graupel and graupel-snowflake collisions under still air conditions at -15 °C and over water saturation. The particles were synthetically generated within a cold room through two distinct methods: riming and vapor deposition for graupel with diameters of 2 mm and 4 mm, and by manually sticking vapor grown ice which were generated above a warm bath to form snowflakes with a diameter of 10 mm. All fragments resulting from graupel-graupel collisions were collected and investigated
by means of a digital optical microscope, while fragments from graupel-snowflake collisions were observed and recorded instantly after collision using a holographic instrument. From these experiments, distributions were obtained for fragment sizes, cross sectional areas and aspect ratios. The results showed a higher number of fragments at lower kinetic energy compared to those presented in the literature. 150 to 600 fragments were observed for graupel-graupel with dendrites collisions, and 70 to 500 fragments for graupel-snowflake collisions for collision kinetic energies between $10^{-7}$ and $10^{-5}$ J. Parametrizations for
fragment size distributions are provided with a mode at 75 μm for graupel-graupel with dendrites collisions and at 400 μm for graupel-snowflake collisions. We also propose new coefficients fitted on our experiments to parameterize the number of fragments generated by collisions based on the theoretical formulation of Phillips et al. (2017). These results can be used to improve the representation of ice collision breakup in microphysical schemes.

 # 1 Introduction

The microphysical properties of ice particles within mixed phase clouds play a crucial role in numerical cloud models when determining the amount of surface precipitation (Heymsfield et al., 2020; Field and Heymsfield, 2015) and its temporal evolution (Flossmann and Wobrock, 2010). A particularly well known problem is the observed discrepancies between the concentration of ice nucleating particles and ice crystals (Hallett et al., 1978; Hobbs et al., 1980; Mossop, 1985). This suggests that besides the
primary ice generation processes of heterogeneous and homogeneous nucleation, other processes are also involved. Currently, there is a scientific consensus that secondary ice processes (SIPs) are the key mechanisms responsible for the observed discrepancies in ice crystal number concentration (Field et al., 2017). Korolev and Leisner (2020) discussed six SIPs: (1) shattering during droplet freezing, (2) rime splintering (also called Hallett-Mossop process), (3) fragmentation due to ice-ice collision, (4) fragmentation due to thermal shock, (5) fragmentation of sublimating ice crystals, and (6) activation of ice nucleating particles
in transient supersaturation (i.e. high supersaturation zones) around freezing drops. In this paper we investigate the fragmentation breakup of ice crystals due to collision (point 4 in the above list). One of the first ground based observations reporting the presence of broken ice crystals that might have undergone fragmentation due to collision dates back to the seventies (Jiusto and Weickmann, 1973; Hobbs and Farber, 1972). Fragmentation of ice particles was also studied by in situ observations such as Schwarzenboeck et al. (2009) where a significant portion of the broken crystals was attributed to natural fragmentation, or
by Takahashi (1993) where the presence of high ice concentration was observed. Recently, von Terzi et al. (2022) found that radar signatures can be potentially attributed to ice aggregate collisions in the dendritic growth layer where fragile dendrite crystals grow. Despite the evidence of the existence of this process, only a few laboratory experiments have been performed till date (Vardiman, 1978; Griggs and Choularton, 1986; Takahashi et al., 1995), which introduces uncertainty in parametrizations. According to the parametrization of Phillips et al. (2017), for instance, the fragmentation breakup of ice crystals is often considered as the primary contributor of ice concentration in convective clouds in forecast models (Sotiropoulou et al., 2020, 2021; Zhao et al., 2021; Huang et al., 2022; Karalis et al., 2022; Waman et al., 2022; Patade et al., 2022). Alternatively, in the parcel model of (Sullivan et al., 2018), the fragmentation breakup produces as much ice as drop shattering and rime splintering. Nonetheless, numerous important factors such as the number, size, and shape of the fragments, dependency on temperature and ice particle habit are not precisely quantified, to accurately represent the ice fragmentation process in the models.

The first ice-ice collision experiment presented in the literature was carried out by Vardiman (1978), who observed freely falling natural ice particles colliding with a metal plate. Based on the change of momentum of the incoming ice particles Vardiman (1978) proposed a theoretical model to derive the number of fragments. In this experiment, five types of ice particles have been employed and their collisions were captured by a high-speed movie camera. The collision experiments revealed the production of numerous fragments in the case of light to moderately rimed spatial crystals, a fragmentation dependency on the degree of riming for dendrites, and an ineffective fragmentation of graupel particles. Based on these observations and mathematical model, Vardiman (1978) suggested that the collision between heavily rimed dendrite and graupel can produce ice fragments of significant amount. However, according to Phillips et al. (2017), these experiments could have been affected by sublimation weakening. Furthermore, Korolev and Leisner (2020) pointed out that rotational energy should be considered

for collisions. This is not the case in Vardiman (1978) where a fixed target was used, which may overestimate the number of generated fragments. Nevertheless Phillips et al. (2021) argue that this final rotational energy is just a small fraction of the initial CKE and that this issue can be solved applying Phillips et al. (2017) theory.

Griggs and Choularton (1986) studied the fragmentation of rimed ice and crystals in laboratory. In those experiments, freely falling glass beads of different sizes were made to collide with rimed ice or ice crystals growing on a target rod. The results revealed that a rimed structure does not efficiently produce secondary ice. Conversely, collisions between ice crystals and graupel are very likely produce secondary ice by the fragmentation of the crystal's fragile structure. These conclusions are in agreement with those of Vardiman (1978). Furthermore, spontaneous fragmentation of ice crystals due to aerodynamic drag forces was also observed.

The most comprehensive study was carried out by Takahashi et al. (1995) based on the Takahashi (1993) in situ observation of ice crystals in cumulus clouds. The Takahashi (1993) observations explored a high concentration of 60 and 100 μm ice crystals, when both small (< 2 mm) and large (4 mm) graupels were present. The small graupels were observed having stellar ice branches on their surface. Such graupel with fragile branches were also observed recently by Korolev et al. (2022) at low temperatures. Takahashi (1993) hypothesized that the collision between graupel particles can produce fragments resulting from the breakup of the stellar fragile branches growing on the small graupel surface. Takahashi et al. (1995) aimed to reproduce graupel-graupel collisions in laboratory experiments adopting two ice spheres of 1.8 cm diameter, which were fixed by metal rods as proxies for graupel. While one of the ice spheres was kept stationary to grow by vapor deposition, the other sphere rotated at a speed of 4 ms$^{-1}$ and grew by riming. The two ice spheres were forced to collide with each other, and ice fragments were released. The experiment was repeated for different growth times, temperatures, and collision forces. The results revealed that the number of fragments increases with increasing collision force and time of depositional growth. A maximum of 800 fragments was produced at around -16 °C. This finding supports that the collision of graupel with different types of crystals grown on their surface can produce many secondary ice crystals. However, this laboratory study can be criticized from several aspects. First, the collision kinetic energy (CKE), which is a fundamental quantity that determines the collision outcome of binary particle collisions (e.g. Low and List, 1982; Szakáll et al., 2014; Phillips et al., 2017), was too high in the Takahashi et al. (1995) experiments (Korolev and Leisner, 2020). CKE is calculated as

$$K_0 = \frac{1}{2} \frac{m_1 m_2}{m_1 + m_2} (v_2 - v_1)^2 \qquad (1)$$

where $m_1$ and $m_2$ are the masses, while $v_2$ and $v_1$ are the fall velocities of the large and small graupel, respectively.

Since the mass of the ice spheres of 1.8 cm and their contact area in Takahashi et al. (1995) experiment exceeded by far that of a natural graupel, the CKE resulted in an unnaturally large number of ice crystal fragments as highlighted by Korolev and Leisner (2020). However Phillips et al. (2017) argue that this issue can be fixed using their theoretical scheme for fragmentation during collisions. It should be noted that in that experiment crystals located on the ice sphere were growing on a smooth, non-rimed surface without natural airflow, which does not closely represent the case of a freely falling natural graupel.

Owing to the importance of the process and the lack of reliable laboratory data from experiments, in this paper we introduce a new experimental design to reproduce ice particle collisions more accurately. We also present our first results on the

fragmentation after graupel-graupel (having a vapor grown or bare rimed surface), as well as graupel-snowflake collisions. In our experiments special attention was given to the parameters that might be crucial for the final outcome, namely the fragile structure grown on the ice particle's surface, the collision kinetic energy, and the temperature. Furthermore, the graupels were generated in a way that resembles their growth in the atmosphere, thus, resulting in natural-like particles in terms of size, density, and structure. The experiments were performed inside the Mainz cold room (M-CR) which is a $4\,\mathrm{m} \times 3\,\mathrm{m} \times 3\,\mathrm{m}$ insulated walk-in cold chamber located in the wind tunnel laboratory of the Johannes Gutenberg University of Mainz, Germany. The room temperature was set to -15 °C where fragile dendritic crystals are expected to grow most efficiently.

Our results are rescaled in terms of CKE and ice crystal surface area during collisions using Phillips et al. (2017) theory (Eq. 13 therein) where the number of fragments generated by collisions is expressed by

$$N = \alpha A(M)\left(1 - \exp\left(-\left[\frac{CK_0}{\alpha A(M)}\right]^{\gamma}\right)\right) \tag{2}$$

with $\alpha$ being the surface area of the smallest ice particle, $A(M)$ the number of breakable asperities of the ice crystal per area, $C$ the fragility asperity coefficient, $K_0$ the collision kinetic energy and $\gamma$ the shape parameter.

This paper is organized as follows: in Section 2 the experimental setup and procedures for ice particle generation are described. The setup for ice particle collisions is introduced in section 3, while the results for the fragment number, size and shape are presented and analysed in section 4. After the discussion of the results we also provide a critical overview of the constrains of our experiments.

## 2 Experimental setups for ice particle production

### 2.1 Generation of nature-like graupel particles

Nature-like graupel were generated by means of the GEORG (GEnerator instrument Of Rimed Graupel) apparatus (see details in Theis et al., 2022). This setup consists of a flow tube with a square cross section (17 cm x 17 cm x 200 cm) which is placed inside the M-CR. Inside the flow tube a small epoxy spheroid about $1 \pm 0.2$ mm diameter is placed, mimicking a frozen drop and acting as the embryo for the graupel. The epoxy core is attached to a 40 μm diameter fiber which is fixed to a double gyration apparatus inside the flow tube. This apparatus consists of one electric motor rotating around the vertical axis, and a second one which is fixed at 45° angle with respect to the first one. Both motors are rotating at 4 Hz simulating the rotation and tumbling motion of a freely falling graupel. The whole surface of the epoxy embryo was exposed this way to a supercooled droplet flux in order to produce a lump graupel (classified as Rb4 in Kikuchi et al., 2013). A constant flow speed of $2.78 \pm 0.10\,\mathrm{ms^{-1}}$ was produced by a blower on the top of the flow tube to simulate the free fall of the graupel and to maintain a central droplet stream inside the flow tube. Liquid droplets with a modal diameter of $19.7 \pm 5.1$ μm, arithmetic mean diameter of $23.4 \pm 8.8$ μm and a volume diameter of $34.6 \pm 9.9$ μm were generated by the means of an ultrasonic atomizer (US 2/58 Hz, Lechler GmbH, Germany) which was fed by $15\,\mathrm{mLmin^{-1}}$ input nitrogen gas flux. An ice filter was added at the top of the flow tube to avoid ice accretion inside the blower.

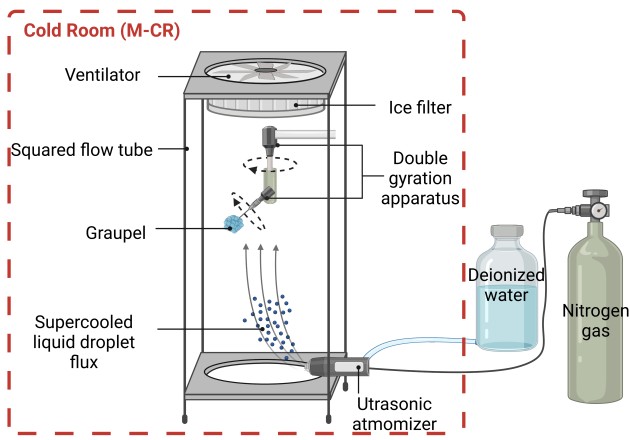

**Figure 1.** The apparatus GEORG used for the generation of nature-like lump graupel (Theis et al., 2022, modified after). The flow tube was placed inside the M-CR at -15 °C, while the gas and water bottles were kept outside the cold room.

To characterize their mass, each graupel was melted and the size of the resulting liquid drop was measured by following the procedure of Theis et al. (2022). Before melting, each graupel diameter was estimated from microscope images adopting three methods: (1) the equivalent area diameter from the graupel's apparent cross section; (2) two equivalent volume diameters from two estimations of the graupel's volume determined by an elliptical fit on the graupel edges contours; (3) two equivalent volume diameters from an integration method used in Szakáll et al. (2014). The resulting statistics of the mean graupel characteristics is given in Table 1.

| Temperature (°C) | LWC ($\mathrm{gm^{-3}}$) | Growth time (s) | Number of graupel | Diameter (mm) | Density ($\mathrm{gcm^{-3}}$) |
|---|---|---|---|---|---|
| -10 ± 1.5 | 3.46 ± 0.23 | 180 | 5 | 2.43 ± 0.3 | 492 ± 88 |
| -10 ± 1.5 | 3.50 ± 0.78 | 420 | 2 | 4.47 ± 0.36 | 782 ± 166 |
| -15 ± 1.5 | 2.32 ± 0.16 | 180 | 5 | 2.24 ± 0.17 | 334 ± 62 |
| -15 ± 1.5 | 2.20 ± 0.12 | 360 | 5 | 3.51 ± 0.36 | 510 ± 70 |
| -20 ± 1.5 | 0.71 ± 0.08 | 420 | 5 | 2.05 ± 0.28 | 248 ± 23 |

**Table 1.** Graupel mean characteristics for different temperatures, growth times and liquid water content (LWC) generated using the GEORG instrument (Fig. 1).

For each graupel, the volume determined by the three previous methods and the mass estimated from the melting were used to calculate its density. For the collision experiments, graupels were generated at -15 ± 1.5 °C under the same conditions as graupel presented in Table 1. The two densities obtained for this temperature are 334 ± 62 $\mathrm{gcm^{-3}}$ for the graupel with roughly 2 mm size and 510 ± 70 $\mathrm{gcm^{-3}}$ for roughly 4 mm graupel (the choice of graupel size is explained in section 3.1.1). These values for the two densities needed to be used for the calculation of the graupel masses and CKE because melting the graupel after the experiments was not possible.

## 2.2 Ice crystal growth by water vapor deposition

Dendritic ice crystals were grown on the surface of graupel particles by deposition of water vapor. For that, a glass aquarium setup (Fig. 2.), similar to the one described by Fries et al. (2006) was utilized in the M-CR. At the top of the aquarium, supersaturation with respect to ice was maintained by humidifying the cold dry ambient air with water vapor evaporated from a heated

water bath. The water vapor saturation level was determined by the M-CR air temperature and by the water bath temperature which was controlled and stabilized using a heating resistance plate. A homogeneous water bath temperature was ensured by the continuous stirring of the water using a magnetic stirrer. To characterize the crystal growth, two PT-100 sensors were mounted inside the aquarium to measure the air and water temperatures, respectively. The ice supersaturation was determined from the dew point measured by a chilled mirror dew point hygrometer (DP3-D/SH, MBW Elektronik AG, Switzerland) in the

immediate vicinity of the growing ice crystals. In order to avoid condensation inside the sampling tube, it was coiled up and heated by a conductive wire dissipating heat.

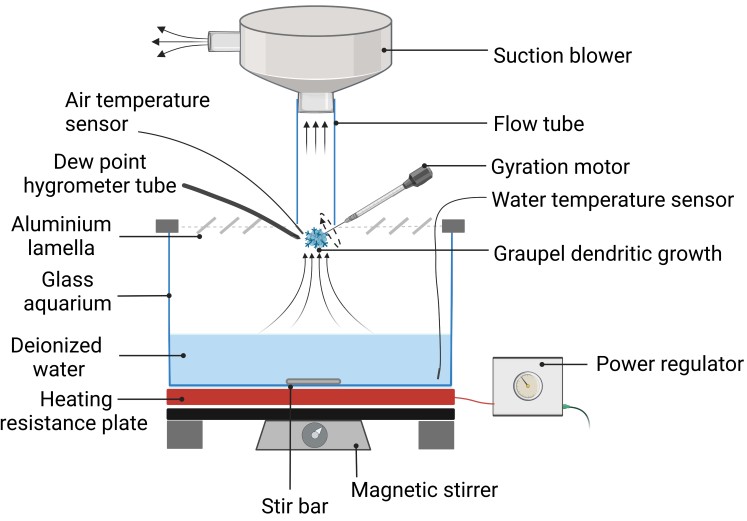

**Figure 2.** Experimental setup for producing dendritic ice crystals on graupels' surface. See the text for details.

To reproduce graupel with vapor grown crystals on their surface as observed by Takahashi (1993), a 2 mm graupel was held by a fibre inside a cannula over the aquarium on a 4 rpm gyration motor. The motor simulates the rotation of a natural graupel

and induces a homogeneous dendritic growth on the graupel surface. A blower and a flow tube were placed above the graupel particle in order to simulate the natural air flow pattern of its fall. The flow speed was set at the position of the graupel to $2.50 \pm 0.25$ ms$^{-1}$ which is in the range of a 2 mm diameter graupel's terminal velocity (Heymsfield and Wright, 2014). The temperature of the M-CR was kept at -20.0 $\pm$ 1.5°C which lead to an air temperature at the position of the graupel between

-13 and -15 °C. The water bath temperature was set to $13.1 \pm 0.2°C$, resulting in a supersaturation over ice between 20 and 27% (Fig. 3).

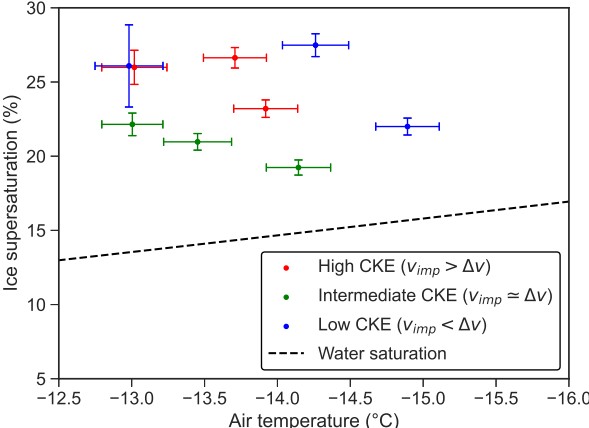

**Figure 3.** Temperature and ice supersaturation during deposition growth for graupel used in collision experiments (see in Section 4.1). The graupel employed for low CKE experiments are plotted in blue, for intermediate CKE in green, while for high CKE in red (see section 3.1.2).

In a natural cloud with $6.6 \times 10^{-3}$ $°Cm^{-1}$ temperature gradient, a 2 mm graupel falling at 2.5 $ms^{-1}$ would experience a temperature increase of 5°C within 5 minutes. Hence, such a graupel would stay in the dendritic growth zone between -20°C and -10°C during this time in a cloud without updraft, as observed in Takahashi (1993). Therefore, we allowed 5 minutes growth time of dendritic crystals in our experiments.

In order to verify the ability of our setup to produce realistic cloud ice crystals, the blower was turned off to produce vapor grown ice crystals in still air conditions. In this case, numerous ice crystals were growing on the aluminum plates at the top of the aquarium. These crystals were collected and investigated using an optical microscope in order to characterize their shape following Kikuchi et al. (2013).

Figure 4 provides a comparison of temperature, ice supersaturation and crystal type obtained in our experiments (symbols) with Kobayashi (1961) diagram which is represented by black solid and dotted lines, with a special focus on the dendritic growth zone around -15°C. The Kobayashi (1961) ice crystal diagram has been assessed and found to be in agreement with other laboratory studies and cloud observations (Pruppacher and Klett, 2010). As demonstrated in Fig. 4 our aquarium setup produces ice crystals similar to those expected in the clouds and that its use is appropriate for collisions experiments.

Figure 5 depicts dendritic ice crystals grown on the surface of an ice sphere and of a graupel particle. The graupel was generated in GEORG at -15°C (see Table 1), while the ice sphere was produced by wet growth, adding successively liquid layers around the ice core and waiting for complete freezing. Dendrites grown on the supporting wires of the two ice particles

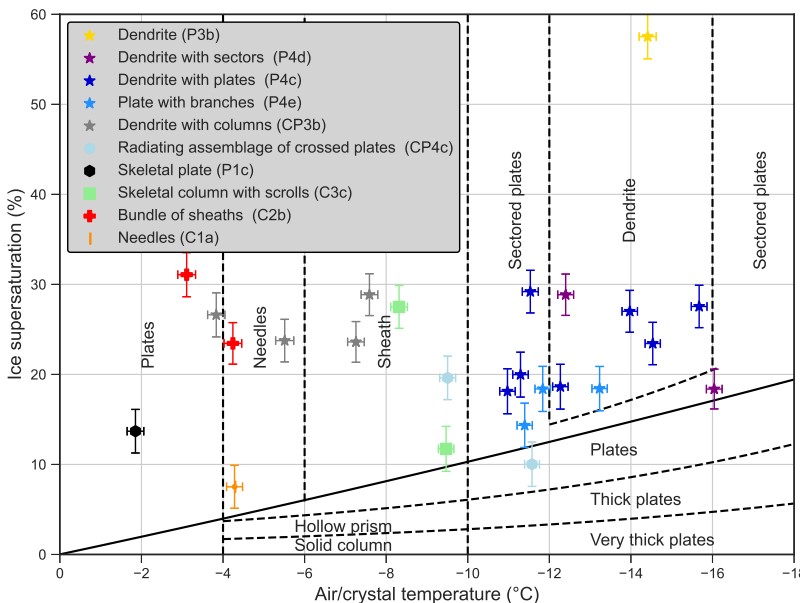

**Figure 4.** Depositional grown ice crystals inside the aquarium setup categorized following Kikuchi et al. (2013) and compared to Kobayashi (1961) diagram.

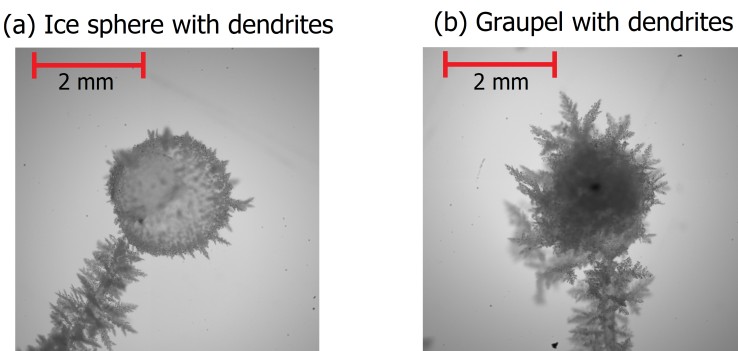

**Figure 5.** Ice sphere with dendritic crystals similar to Takahashi et al. (1995) and graupel with dendrites used in this study for collisions. For both ice particles, dendritic crystals (around 700 μm maximum dimension for the longest one) are produced after 5 minutes of vapor deposition (T ≈ -14 °C and $S_{ice} \approx 23\%$) by means of the aquarium setup shown in Fig. 2

have similar sizes which demonstrates that the growth of crystals are similar under the same conditions and the same surface of deposition. The maximum size of a dendritic crystal observed in Fig. 5a was approximately 400 μm, and 700 μm in Fig. 5b.

The structure of pure ice which was created by wet growth apparently inhibits the growth of dendrites compared to the rough structure of rimed graupel. Since Takahashi et al. (1995) used smooth ice spheres, collision experiments can be different to those where the dendritic growth takes place on rimed (i.e., rougher) surfaces.

## 3 Collision experiments

### 3.1 Graupel-graupel collisions

The purpose of the first series of collision experiments was to reproduce the Takahashi et al. (1995) experiments in a more realistic manner in terms of ice particle size, and to develop an improved method to detect and count the generated ice crystal fragments. Therefore, the decision to select 2 mm and 4 mm graupel sizes for the collision experiments was made based on the particle sizes observed in Takahashi (1993) study, which estimated the expected collision of graupel particles in the atmosphere. Two distinct collisions involving 2 mm and 4 mm graupel particles were carried out following the same procedure. One where dendritic crystals were grown on the surface of the 2 mm graupel (as observed by Takahashi, 1993) and the second one where the 2 mm graupel surface was bare.

#### 3.1.1 Setup for graupel-graupel collision experiments

Graupel-graupel collisions were performed inside a 4.5 cm diameter "collision tube" as shown in Fig. 6. An additional, 8 mm diameter "drop tube" was placed on the top of the "collision tube" to guide the falling graupel to the impact point. A 2 mm graupel with vapor grown crystals on its surface (or with a bare surface) was carefully and quickly moved from the aquarium to its proper position in the "collision tube". It was held by a thin wire fitted into a cannula. As this wire possesses only a small aerodynamic resistance, the graupel was allowed to move after collision (see Fig. 7). The 4 mm graupel's supporting wire used for its generation was cut close to its surface and held with tweezers before launching it through the "collision tube". An example of a video recorded during an experiment on graupel-graupel with dendrites collision is provided as a supplement of this paper (see https://doi.org/10.5446/62064).

The setup was placed over the glass aquarium to avoid any sublimation weakening of the dendritic crystals. Each collision was captured by a high-speed camera (MotionPro Y3) at 1000 fps. A petri dish filled with a thin layer of paraffin oil ($\rho \approx 0.85 \pm 0.03 \ \mathrm{gcm}^{-3}$) was placed at the bottom of the collision tube to collect the ice fragments generated by the collision. A few seconds after the collision, the petri dish was covered by another layer of paraffin oil previously stored at -5°C. As the cold room temperature was around -20°C, the oil solidified which immobilized the crystals on the petri dish. After being collected, pictures of fragments were taken and later analysed (see section 3.1.3).

#### 3.1.2 Fall speed and collision kinetic energy

The small graupel with dendrites was fixed in the "collision tube", resulting in a reduction of the terminal velocity difference expressed in the original CKE equation (Phillips et al., 2017) to the fall speed of the large graupel $v_1$ in Eq. 1. As a result, the

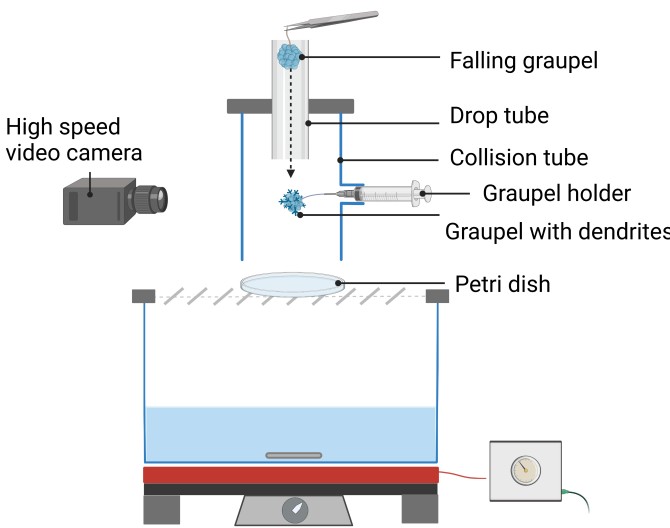

**Figure 6.** Experimental setup for graupel-graupel collisions over the glass aquarium inside the M-CR

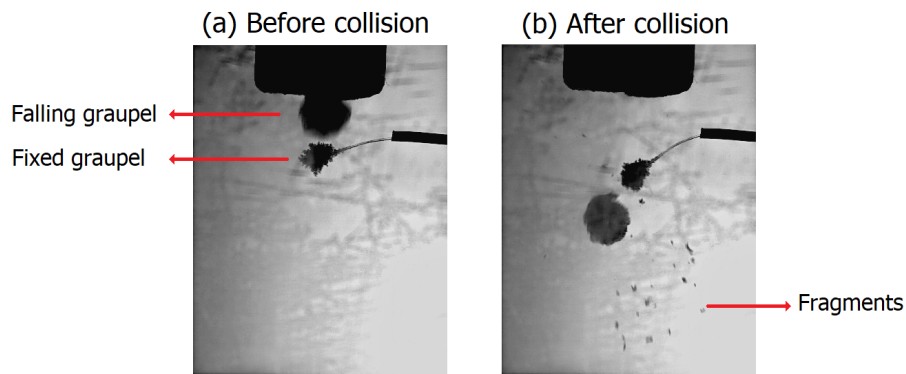

**Figure 7.** Images of graupel in the collision tube (a) before and (b) after collisions as captured by a high speed camera

CKE is determined by the masses of the graupel particles and the velocity $v_1$ of the large graupel. However, in natural clouds, both particles would be falling, which implies that $v_1$ should be equal to the terminal velocity difference of the two graupel particles $\Delta v$ for our experiment.

From Eq. A9 in Appendix A and a realistic drag coefficient $C_d$=1 for graupel of the investigated sizes (Heymsfield et al., 2018), the two theoretical graupel terminal velocities are $v_T(D = 4 \text{ mm}, \rho_g = 0.51 \text{ gcm}^{-3}) = 4.36 \text{ m.s}^{-1}$, and $v_T(D = 2 \text{ mm}, \rho_g = 0.34 \text{ gcm}^{-3}) = 2.50 \text{ ms}^{-1}$. The typical terminal velocity difference between these two graupel is therefore $\Delta v$=1.86 ms$^{-1}$. From this typical terminal velocity difference, three tube lengths were chosen (Table 2) depending on the large graupel's

velocity just before the collision as expressed in Eqs. A7 and A8. By employing three different falling distances, one can modify the collision kinetic energy and observe how it impacts the outcomes of collisions. The small tube corresponds to an impact speed $v_1 < \Delta v$, the intermediate tube to $v_1 \simeq \Delta v$ and the long tube to $v_1 > \Delta v$. The three CKE ranges in Table 2 will be

| Tube | Tube length (cm) | Calculated $v_1$ (ms$^{-1}$) | Measured $v_1$ (ms$^{-1}$) | Calculated CKE (J) | Measured CKE (J) |
|---|---|---|---|---|---|
| Short tube | 5 | 1.14 | $1.02 \pm 0.15$ | $8.4 \times 10^{-7}$ | $1.2 \times 10^{-6} \pm 0.4 \times 10^{-6}$ |
| Intermediate tube | 22 | 1.86 | $1.96 \pm 0.15$ | $2.2 \times 10^{-6}$ | $4.7 \times 10^{-6} \pm 1.3 \times 10^{-6}$ |
| Long tube | 80 | 3.58 | $3.06 \pm 0.15$ | $8.3 \times 10^{-6}$ | $1.6 \times 10^{-5} \pm 0.4 \times 10^{-5}$ |

**Table 2.** Mean graupel fall speeds and CKEs measured (from collisions experiments) or calculated for different collision tube lengths adopting Eqs. A7 and A8. The tube lengths are corresponding to velocity differences of a 2 mm ($\rho = 334 \pm 62$ gcm$^{-3}$) and a 4 mm ($\rho = 510 \pm 72$ gcm$^{-3}$) graupel.


referred hereinafter as low, intermediate and high CKE. Graupel-graupel with dendrites collisions were carried out 3 times for each kinetic energy range. For each collision, the CKE was calculated under the above mentioned consideration. Since in our setup it was not possible to melt the graupel after the collisions, their masses were derived from the mean densities from the characterization measurements at -15 °C (see Table 1), and from their volume calculated from their images described in section

2.1. The mass of vapor grown dendritic crystals on the small graupel was neglected. Images recorded by the high speed camera were used to calculate the large graupel's fall speed $v_1$ with an accuracy of 0.15 ms$^{-1}$ according to the camera pixel size and frame rate.

### 3.1.3 Crystal image processing

The ice crystal fragments inside the petri dish were analyzed unsing a microscope. For that, the petri dish surface was subdi-

vided into a 5mm × 5mm grid (see Fig. 8a). Microscope pictures of 3.00 μm pixel resolution were taken for each grid cell. Because of the numerous dust particles present in the petri dish, ice fragments were identified manually and contoured on the computer image by red rectangles as shown in Fig. 8b. For each picture, the portions delimited by red rectangles in each grid mesh were selected and counted as a single ice particle (Fig. 8c).

A double gradient, which consists of summing the intensity gradient of rows and columns of the image pixels, was applied

to identify crystal outlines. The gradient pictures were binarized using a threshold value chosen and applied for all images and fragments. Dilatation-and-erosion as morphological closure was applied to fill the holes in the dendritic structure images and get a single crystal. Dust particles inside the red rectangles around the chosen crystals (see Fig. 8c) were automatically filtered out by considering only the largest particle cross sectional area which are ice fragments (see Fig. 8d). From the individual binarized fragments, the major and minor axes were obtained by fitting an ellipse in the same way as for graupel particles (see

Section 2.1).

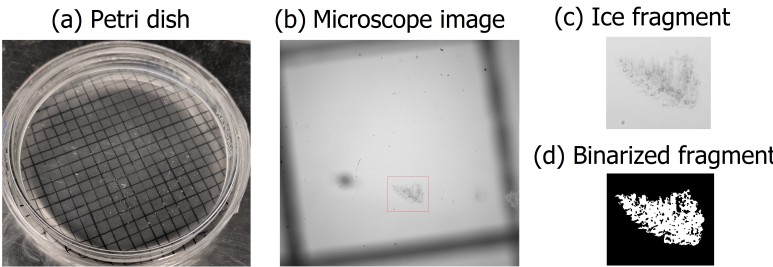

**Figure 8.** (a) Ice fragments collected after a graupel-graupel collision on the petri dish filled with paraffin oil. (b) Microscope picture of ice fragments inside a grid cell. Image processing steps are shown in (c) and (d).

## 3.2 Graupel-snowflake collisions

Snow crystal aggregates (called snowflakes hereinafter) around $10 \pm 2 \, \mathrm{mm}$ were manually formed by sticking (i.e. by interlocking) branches of the dendrites together around 20 to 60 millimeter-size P3b/P4d ice crystal monomers. These dendritic crystals were grown inside the glass aquarium in still air (Fig. 2) using the temperature and humidity conditions provided in
Fig. 4 for P3b/P4d crystals. Although the microscope method is utilized to identify fragments in graupel-graupel collisions, it is not suitable for graupel-snowflake collisions due to the overlapping of dendrite fragments in the petri dish, which obstructs their accurate identification. To overcome this issue, the "Holographic Imaging and Velocimetry Instrument for Small Cloud Ice" (HIVIS) Weitzel et al. (2020) was introduced to detect the ice fragments produced by freely falling graupel-snowflake collisions. Consequently, holographic images were captured for the ice fragments generated from graupel-snowflake collisions.

### 3.2.1 Setup for graupel-snowflake collisions

The collision took place in a transparent tube of $1.9 \, \mathrm{cm}$ inner diameter and $50 \, \mathrm{cm}$ length placed above the HIVIS instrument as shown in Fig. 9. A snowflake was placed onto a $1.3 \, \mathrm{cm} \times 1.3 \, \mathrm{cm}$ squared plate mounted on a hinge. Next, the snowflake was launched by turning the hinge, and at the same time, the graupel was starting to fall from a few centimeters above the tube. Since the fall speed of the graupel is higher than that of the snowflake, the collision took place as soon as the graupel
overtook the snowflake. As the launch of the snowflake was carried out by the operator, the place where the graupel caught the snowflake was unpredictable. Hence, collisions had different collision kinetic energies depending on the actual velocity difference of the particles at the instant of the collision. Video sequences of central and edge collisions are shown in Fig. 10, while the complete videos of collisions are provided as supplements of this paper (see https://doi.org/10.5446/62066 and https://doi.org/10.5446/62065).

After the collision, all fragments were falling through the tube toward the sampling volume of the HIVIS instrument and crossed the laser beam. Ice fragment falling through a $2.2 \times 2.2 \times 4 \, \mathrm{cm}$ sampling volume were captured by a camera (Basler Ace 2) at 90 fps with a pixel resolution of $10.87 \, \mu\mathrm{m}$. All the fragments were collected in a Petri dish located sufficiently at the bottom of the HIVIS to avoid interfering with the detection of freely falling fragments in the laser sampling volume. The

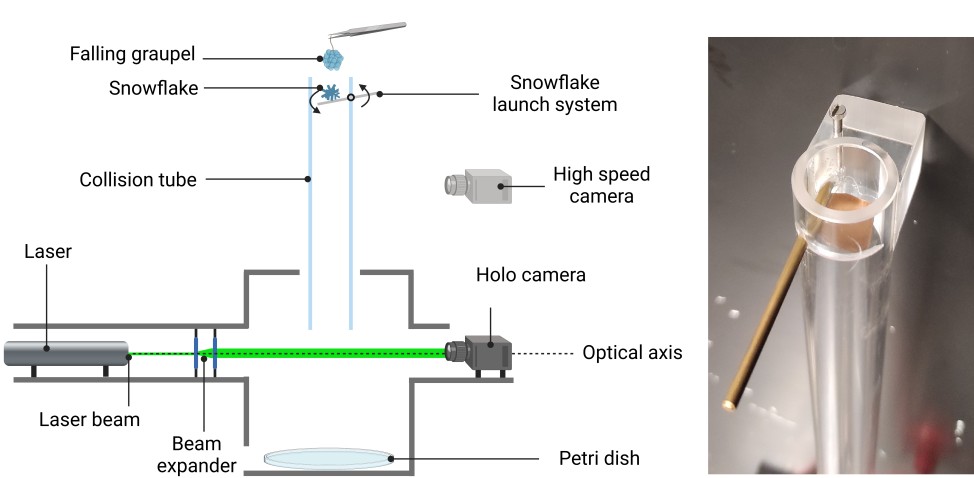

**Figure 9.** (a) The HIVIS instrument (Weitzel et al., 2020) for graupel-snowflakes collisions inside the M-CR composed and (b) the snowflake launching system.

fragments were then melted and weighted to measure the total snowflake mass. The CKE was calculated from this mass and the
images recorded at 400 Hz by the high speed camera, which imaged the entire tube. The graupel could collide with a snowflake at different positions measured from the snowflake's center, which might influence the generated fragment number or size. In order to investigate this effect, the impact position parameter $I_p$ similar to eccentricity for drop-drop collisions (see Szakáll and Urbich, 2018) is introduced as

$$I_p = 1 - \frac{x_g - x_s}{R_s} \tag{3}$$

where $x_g$ and $x_s$ are the apparent horizontal center position of the graupel and the snowflake, respectively, and $R_s$ is the radius of the snowflake calculated as half of the snowflake maximum dimension. In case of central collision $I_p = 1$ (Fig. 10a), while for edge collision $I_p = 0$ (Fig. 10b). This parameter is calculated from the pictures of only one camera point of view which imposes limits to the precision of $I_p$.

As snowflakes were formed under still air conditions, some crystal monomers or crystal branches were expelled because of
the increasing of drag forces during the fall. To characterize this artificial enhancement of ice crystal fragment number in the results, five snowflakes were launched into the tube without collision. This blank measurement resulted in 26 to 140 fragments, with a mean of 51 fragments. This number of fragments was subtracted from the total number of crystals produced by collisions. It should also be noticed that these collisions were carried out in undersaturated conditions around $90\%$ ice saturation. Although sublimation can potentially cause dendritic structures to weaken, resulting in an increased production of fragments compared to
saturated conditions, this effect was likely limited in this experiment. This is because the snowflake spent less than 30 seconds

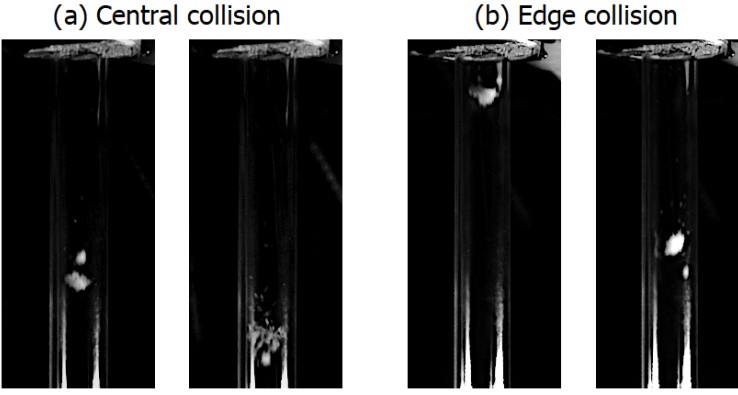

**Figure 10.** Two types of graupel-snowflake collision above the HIVIS instrument inside the cold room with (a) central collision and (b) edge collision. The left picture represents the two particles before and right picture after the collision for (a) and (b).

in the air under these conditions before falling, and the fall of both the snowflake and its fragments lasted only about 1 second. As a result, any sublimation-induced weakening effect on the size and number of fragments observed was likely minimal.

### 3.2.2 Hologram reconstruction and fragment tracking

Crystal images were reconstructed from holograms (see details in Weitzel et al., 2020) at each $\Delta z$=100 µm distance along the optical axis (see Fig. 9) using the method of Fugal et al. (2004). The particle properties were determined in terms of the major and minor axes. Furthermore, the cross sectional area was obtained from a particle detection algorithm of Fugal et al. (2009). To distinguish ice fragments and artifacts (i.e. holograms resulting from noise, optics imperfection and reconstruction method), a decision tree based on the particle properties was created and applied to the reconstructed objects for each collision.

As fragments are falling slowly through the sampling volume, the same crystal can be seen on successive holographic images. To avoid counting crystals several times and to identify the repetition of the same fragments between frames, a tracking program was developed. The tracking was based on predicted position of the fragments at the subsequent time step from an initial fragment position. This prediction is based on the fall speed parametrization $v_T = 0.67 D_{max}{}^{0.46}$ (Vázquez-Martín et al., 2021) valid for spatial stellar ice crystals. All fragments present in the area of an ellipse surrounding the predicted position were selected as being the potentially preceding crystal. During the analysis, the ellipse size was manually adjusted to 6 mm major axis and 1.2 mm minor axis to be able to track all crystals. The size of the ellipse was chosen after several tests to follow the large millimeter sized crystals while avoiding counting other nearby crystals. The dimensions (maximum size,cross-sectional area, and aspect ratio) of all fragments found in the designated area were compared to those of the initial crystal. If one of these dimensions varied by more than 30% from the preceding particle, the fragment was not considered to be the initial one. After multiple comparisons with a visual track of the fragments, this deviation was deemed appropriate and provides the best balance for identifying the correct fragments. This condition is especially useful when multiple fragments are present in the ellipse area and helps to select a crystal that matches the characteristics. The program was also tested using deviation values

of 20% and 60% to determine the error in the total number of fragments produced (see error bars in Fig. 14), accounting for factors such as rotation of the fragments.

# 4 Results and discussion

## 4.1 Graupel-graupel collisions

The measured number of fragments produced by graupel-graupel collisions from our experiments as a function of the CKE is shown in Fig. 11 for graupel-graupel with dendrites and bare graupel-graupel collisions.

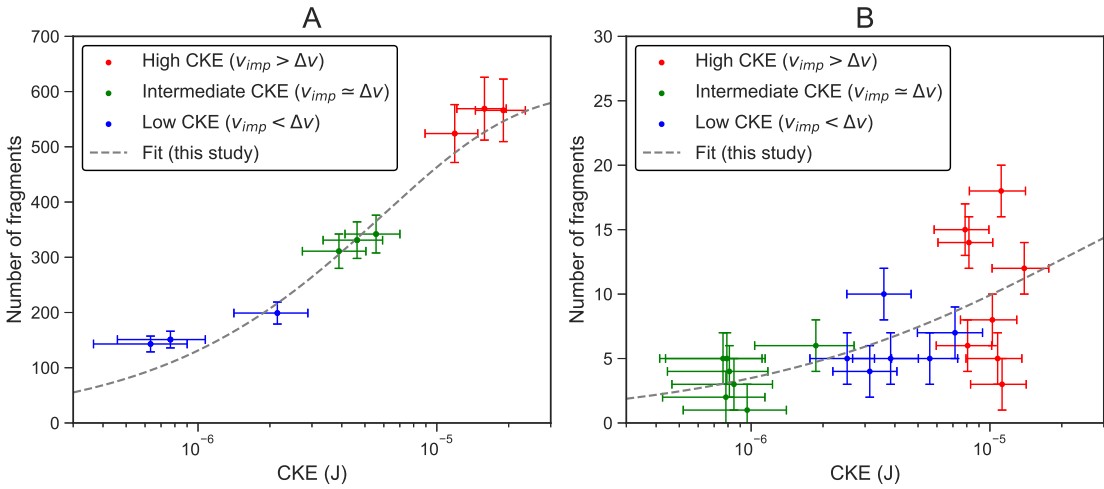

**Figure 11.** Number of fragments (A) produced by graupel-graupel with dendrites collisions at high supersaturation ($S_{ice} \approx 23$ %) and (B) produced by bare graupel-graupel collisions as a function of the CKE. Low CKE in blue, intermediate in green, and high in red (see section 3.1.2). The dashed lines correspond to the fit of the experiments on Phillips et al. (2017) formulation (see Eq.2)

Apparently for both types of collisions, the number of fragments increases exponentially with the CKE which was expected regarding Takahashi et al. (1995). The fit with Eq.2, which is from the theoretical formulation of Phillips et al. (2017), is represented by a dashed line in Fig. 11 and gives the values $A(M) = 1.9 \times 10^8$ m$^{-2}$, $C = 10^8$ J$^{-1}$ and $\gamma = 0.78$. Apparently, the theoretical formulation fits our experimental data well after re-evaluating the coefficient of Eq. 2 by applying the least squares method and assuming that for kinetic energies exceeding CKE $> 10^{-4}$ J, the fragment number is equal to the maximum number of fragments observed (i.e. 569 for graupel-graupel collisions with dendrites, and 20 for bare graupel-graupel collisions). It is important to note, that the accuracy of these coefficients for predicting fragment number become uncertain beyond the experimentally measured CKE values.

It is expected that a maximum number of ice fragments is reached at a certain CKE regarding (Takahashi et al., 1995) experiments and (Phillips et al., 2017) theory. Such a maximum is not observed here within the investigated CKE range between

$10^{-7}$ J and $2\times10^{-5}$ J. Using the original parameterization of Phillips et al. (2017) which is based on the data of Takahashi et al. (1995), a fragment number of less than 20 would be predicted, i.e. much less than in our measurements for graupel-graupel with dendrites collisions but close to bare graupel-graupel collisions. Or, stated differently, a higher CKE between $5 \times 10^{-4}$ J and $5 \times 10^{-3}$ J is required in Phillips et al. (2017) to generate a fragment number between 200 and 450 for hail-hail collisions. Considering the intermediate CKE which represents typical collision conditions in natural clouds (Takahashi, 1993), an average of 310 fragments were ejected during graupel-graupel with dendrites collisions. This is in contrast to Takahashi et al. (1995) who estimated 60 fragments from their experiments. This difference could be due to the use of two 1.8 cm rigidly fixed ice spheres instead of millimeter-size flexible mounted rimed graupel, as in our experiments. As illustrated in Fig. 2, the pure ice structure reduces vapor growth of ice crystals compared to a rimed graupel surface. Furthermore, Takahashi et al. (1995) experiments did not take into account the ventilation effect of a falling graupel, which can, due to its cooling effect, increase the growth rate of the crystals. These two possible effects imply that the Takahashi et al. (1995) experiments very likely lead to a slow crystal growth, which can of course induce a reduction of the number of fragments as the fragility of ice crystals increases with their size (Phillips et al., 2017). We also observed that for longer vapor deposition growth times, the number of fragments increases similarly to the finding in Takahashi et al. (1995). Up to 800 fragments were produced by graupel-graupel collisions in our study for 20 minutes growth time, which is close to the maximum number of crystals observed at -15°C in Takahashi et al. (1995). The mean number of the crystals growing on the 2 mm graupel surface was estimated to be around 38 $\pm$ 13 from microscope images (see Fig. 5). This number is probably underestimated since crystals smaller than 100 µm were not identifiable on the graupel surface. The average size distribution of these crystals is represented in Fig. 12 by a black dotted line.

The growth of dendrites on the graupel surface that occurs under high supersaturation conditions is faster than at low supersaturation, and therefore, may result in a more fragile ice crystal structure. This might lead to more fragments produced by graupel-graupel with dendrites collisions compared to ice crystals growing at lower humidity. Cloud graupel may experience several growth processes that influence their surface properties, making their fragility dependent on their growth history. Consequently, graupel collisions of the same size, with the same collision kinetic energy, can yield different fragment numbers due to their distinct surface properties. Since the results and parameters from Eq. 2 are obtained under high humidity around -14 °C, caution in their use is essential as they only correspond to the specific environmental conditions of our experiments. To further explore the effect of graupel surface properties on fragmentation by collision, rescaling the results (i.e., varying parameters from Eq. 2 based on temperature, humidity, and growth history) would be interesting. However, further experiments should be performed since only Takahashi et al. (1995) studied the effect of temperature on the number of fragments produced by collisions. The previous remarks concerning the surface properties of graupel are supported by the results of bare graupel-graupel collisions which revealed that the number of fragments produced is negligible compared to graupel-graupel with dendrites collision. This agrees with the finding in Griggs and Choularton (1986) where a pure rime structure was inefficient to produce ice fragments.

For each of the graupel-graupel with dendrites collisions, a fragment size distribution (FSD) was calculated from the analysis of the ice crystal images. The average FSDs for the three CKE ranges are presented in Fig. 12. The FSD obtained from the single experiments can be found in Appendix B1. Regarding the low number of fragments obtained for bare graupel-graupel collisions the FSDs are not represented here.

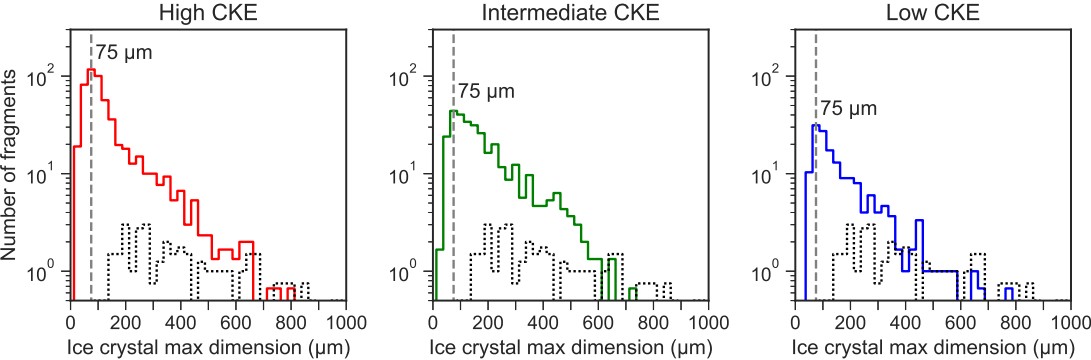

**Figure 12.** Average fragment size distributions as solid line for different CKE ranges of graupel-graupel with dendrites collisions at high supersaturation ($S_{ice} \approx 23\,\%$): blue: low; green: intermediate; and red: high CKE. Average size distribution of crystals on graupel surface as black dotted line.

As obvious from Fig. 12, the FSDs have similar shapes, all having a maximum of fragment number around 75 µm. This result is in agreement with Takahashi (1993) in-situ observation of 60 and 100 µm crystals at -15 °C and -20 °C, respectively, when both large and small graupel were present. Our laboratory study supports therefore Takahashi's hypothesis, who suggested that the observed airborne ice crystals were produced by graupel-graupel collisions. A general parametrization of the FSD for all CKEs is given in Fig. 13.

Similarly to drop-drop collisions of Low and List (1982), a log-normal probability density function is used to fit the collision experiments:

$$f(D) = \frac{1}{\sigma D \sqrt{2\pi}} e^{\frac{(\ln(D)-\mu)^2}{2\sigma^2}} \tag{4}$$

with $D$ the fragment maximum dimension, $\mu$ the position parameter and $\sigma$ the shape parameter. The two parameters $\mu$=4.86 and $\sigma$=0.69 are obtained from all collisions experiments, considering 3136 fragments. These values can be considered as valid for our CKE range at -15 °C. However, in a first instance, we suggest that the parameters $\mu$ and $\sigma$ of the FSD can be interpolated/rescaled considering the size of the parent particle (2 mm graupel here and 10 mm snowflake in section 4.2).

In case of discrete bins, such as in Fig. 13, a simple multiplication with the bin width is necessary to get the crystal size probability with $P(D_i) = f(Di)\Delta Di$. A higher CKE increases slightly the probability to get small fragments (see Fig. 12). However, this effect is restricted to small crystals around 25 µm and remains negligible for the FSD parametrization.

Area and aspect ratio distributions of the fragments were also obtained from the microscope images and are presented in Fig. C1 and Fig. D1. The shape of the area size distributions seems to be independent of the CKE. However, the minimal

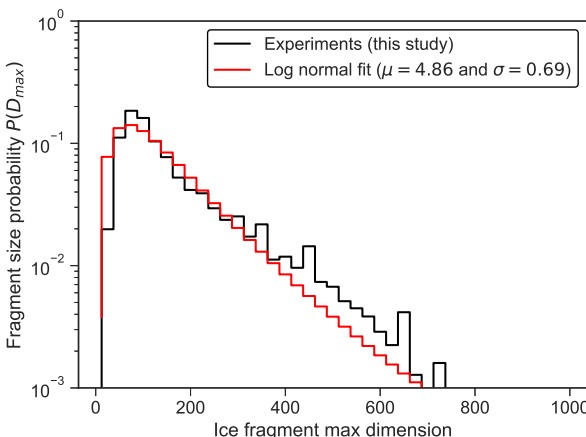

**Figure 13.** Parameterization for fragment size distribution from our collision experiments. Integrated log-normal probability density function in red and initial data from experiments in black.

surface area of the fragments is located around $2 \times 10^{-4}$ mm$^2$ for high CKE, while it is $5 \times 10^{-4}$ mm$^2$ for intermediate and low CKE. This decrease of the minimum fragment area can be explained by the fact that the work done to break crystals is proportional to their cross-sectional area (Phillips et al., 2017). This consideration also applies to the observed higher frequency of 25 μm crystals as the CKE increases (see Fig. 12). However, this observation should be taken with caution because the limit of detection of ice fragments with the microscope is estimated to be around 20 μm for crystal maximum dimension and around $10^{-4}$ mm$^2$ for crystal area. Figure D1 depicts the aspect ratio (AR) of the ice fragments, which is calculated as the ratio between the minor and the major length (AR=$D_{min}/D_{maj}$) of the ellipse fitted on the fragments edges. More than 90% of the crystals have an AR higher than 0.4, and crystals with an AR around 0.7 are the most frequent.

### 4.2 Graupel-snowflake collisions

Figure 14 shows the number of fragments produced by graupel-snowflake collisions as a function of the CKE for different impact position parameters. Two distinct cases of collisions were observed: (i) the graupel impacts the center of the snowflake which is therefore completely broken (Fig. 10a); and (ii) the graupel hits the edge of the snowflake breaking some crystals and producing a rotation of the entire snowflake (Fig. 10b). It is apparent from Fig. 14 that the position where the graupel hits the snowflake, influences the number of produced fragments. Less than 200 fragments are generated for $I_p$ close to 0 while more than 200 fragments are produced for $I_p$ close to 1. Collisions with intermediate impact parameters are represented in green/light blue in Fig. 14.

The number of fragments produced by graupel-snowflake collisions is increasing from around 50 to more than 400 depending on the CKE. In Fig. 14, the errors in the number of fragments are calculated as described in section 3.2.2. The Phillips

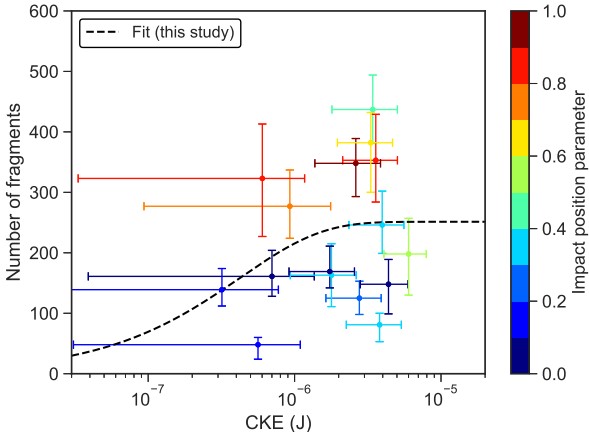

**Figure 14.** Number of fragments produced by graupel-snowflake collisions as a function of the CKE and impact position parameter $I_p$ represented in a color scale from blue to red. The dashed line corresponds to the fit of the experiments on Phillips et al. (2017) formulation (see Eq.2)

et al. (2017) parametrization (see Eq.2) with our re-fitted coefficient is plotted as a dashed line in Fig. 14. The data points corresponding to the central position of the graupel-snowflake collisions are lying slightly above this line while those of edge collisions are lying below. The fit parameters were determined using the least square method with no additional assumptions, and resulted in $A(M) = 2 \times 10^7$ m$^{-2}$, $C = 5.9 \times 10^8$ J$^{-1}$ and $\gamma = 0.78$.

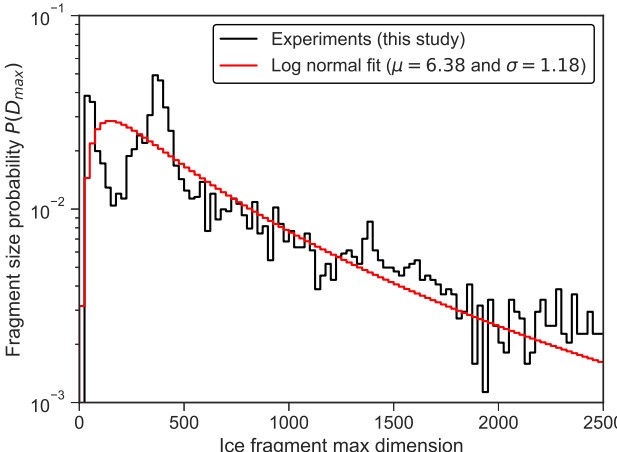

**Figure 15.** Mean fragment size distribution from all graupel-snowflake collision experiments. Integrated log-normal probability density function in red and initial data from experiments in black.

In Fig. 15 the mean fragment size distribution is represented from 16 single graupel-snowflake collisions (see Appendix B2). The majority of fragments are distributed between 0 and 2000 μm with two modes observed around 50 μm and 400 μm. The observed millimeter-size fragments are probably single crystals that are ejected from the snowflake structure during collision while hundred-micrometer fragments might originate from the breakup of single crystals by the collision with the graupel.

Two modes can be identified in the mean FSD of Fig. 15 which is different to the unimodal FSD observed for graupel-graupel collisions in Fig. 13. However, as shown in Appendix B2, these two modes are not systematically present in single collision experiments, in which only half of the collisions resulted in two modes. The presence of these two peaks does not seem to be related to the impact position or to the CKE. It is very likely that the manual production of snowflakes generated diverse ice structures for the different snowflakes that could lead to distinct fragment sizes, and therefore, to the presence of a second mode around 50 μm for some collisions. Since only 16 FSD are presented here, more experiments have to be done to clarify the observation mentioned before. Thus, a general lognormal distribution was fitted to the mean particle size distribution (Fig. 15) from Eq. 4 as for the graupel-graupel with dendrites FSD. This fit with the two parameters $\mu = 6.38$ and $\sigma = 1.18$ can be used to represent the general trend of the distribution but remains imprecise in representing the two modes.

Up to date only Vardiman (1978) carried out ice particle collision experiments with rimed dendrites and graupel. From experimental results and with a model based on change of momentum between the incoming ice particles, he described the time evolution of size distribution from several graupel-rimed-dendrite collisions. The graupel collisions with rimed dendrites (both particles are around 1 mm in diameter) seems to be comparable to our graupel-snowflake collision as our snowflakes were generated using dendritic crystals. In Vardiman (1978), the size of fragments resulting from dendrites were distributed between 0 and 2500 μm, with a mode between 300 and 500 μm which is close to our observations for unimodal distributions for graupel-snowflake collisions. However, no second mode near 50 μm similar to the one observed during our experiments was reported by Vardiman (1978). This can be due to the detection method used in Vardiman (1978) which was probably not able to detect such small fragments, or due to the different particle and experimental setup used for collisions.

For graupel-snowflake collisions the fragment areas are distributed between $5 \times 10^{-3}$ and $10^2$ mm$^2$ (see Appendix C2). It can be noticed that the smallest fragment area of these distribution and the shape of the distributions are close to those of graupel-graupel with dendrites collisions. As can be seen in Appendix D2, the aspect ratio of the fragments in graupel-snowflake collisions is slightly lower than that of graupel-graupel with dendrites collisions, with a maximum of fragments emitted at AR=0.5.

## 5  Limitations of the experiments

The main goal of our experiments was to extend the data of secondary ice production after collision of ice particles, with special focus on particles with dendritic crystals. We intended to simulate atmospheric ice particles in terms of size, morphology, fragility and fall speed. As all laboratory work, this study has also some constrains, which we list and discuss in this section. The listed limitations can be considered in further modelling studies, and can be adressed in future laboratory experiments.

– Artificially generated graupel and snowflakes used in the collision experiments were chosen as proxy for atmospheric particles that might play important roles in secondary ice production processes in clouds. Whether these artificial particles represent significant atmospheric particles may be debated. But considering that if a graupel falls through the dendritic growth zone of a cloud, the environmental conditions are appropriate for dendritic crystals to grow on the graupel's surface. Such dendritic crystals are fragile, and might generate numerous fragments when graupels collide. The same holds true for snowflakes consisting of dendritic crystals. Processes including snowflakes might be more relevant for secondary ice production in the atmosphere than those with graupel, but we aimed to address the investigation with both particle types. Furthermore, investigating graupel was motivated by the Takahashi et al. (1995) experiment, but making a step towards a better representation of atmospheric particles.

– In order to provide a better overview for the significance of the collision of ice particles in secondary ice processes, all measurements should be carried out under different environmental conditions, i.e. at different temperatures and relative humidities. Also a range of ice crystal types in terms of size, rime fraction and morphology should be investigated. Ideally, ice particles should be generated and grown under varying conditions that simulate those inside a cloud. Certainly, realization of such a complex process in a laboratory is impossible. As a first step, we carried out the experiments at one fixed temperature (-15 °C), and the dendrites were grown at around 120% relative humidity with respect to ice.

– The dendritic crystals were grown at a relatively high supersaturation of 20%, which is not typical for clouds. The high supersaturation can modify the structure of the dendrites, and, therefore, their fragility. Hence, our results probably overestimate the number of ice fragments that would be generated in clouds. Nevertheless, the accurate control of humidity levels to generate ice crystals under specific conditions revealed to be a challenging task.

– The size of the dendritic crystals grown on the surface of the graupel, and used also for generating snowflakes was typically 200 µm. The surface density of detectable asperities on the 2 mm graupel surface was around 3 per $mm^2$.

– The size of the snowflakes in our experiments (1 cm) was close to the largest aggregates that can be found in the literature. The fragility of the snowflakes and the number of monomers near the collision path of the incident graupel very likely increase with snowflake size. Thus, our experimental results likely overestimate the number of fragments generated from snowflake-graupel collision in clouds. While the study of collisions of smaller ice particles seems to be more relevant due to their abundance in clouds, it would introduce an additional layer of complexity to make them collide, especially when accounting for free fall experiments.

– The detection limit of our measurements for small ice fragments was approximately 25 to 30 µm. Hence, it is possible that a population of small ice fragment remained undetected.

– It should be noted that keeping the small graupel in position with a fibre creates aerodynamic resistance which can increase the number of generated fragments. However, it is difficult to produce free-fall collisions at the terminal fall speeds of these particles because of their high terminal velocities. Keeping the graupel on a thin wire seems to be the best way to control this experiment while letting the particles freely move after collision.

Advancements in experimental methodologies will offer the potential to address some of these challenges, particularly regarding the augmentation of attainable collision number and the control of the growth conditions of ice particles.

## 6    Conclusions

Two new experimental setups were designed and implemented in the cold room of the Johannes Gutenberg University of Mainz,
Germany, to obtain the number, size and shape of fragments resulting from atmospheric ice particle collisions around -15°C. First, the collision of two graupel particles was studied. In these experiments, a graupel particle with a diameter of 4 mm was released to fall onto another graupel particle with a diameter of 2 mm that had a vapor grown or bare rimed surface. This second particle was mounted on a thin wire, enabling it to move following the collision. All fragments generated during the collisions were collected and investigated under a microscope. The results of graupel-graupel with dendrites collisions revealed that the
number of fragment (150 to 550) is exponentially increasing with the collision kinetic energy (CKE). This evolution and the number of fragments are similar to the findings of Takahashi et al. (1995). However, a significantly higher CKE is required in Takahashi et al. (1995) or Phillips et al. (2017) to obtain the same number of fragments that we found in our experiments. A large number of fragments had a maximum dimension of around 75 μm which is consistent with the 60 to 100 μm ice crystals observed in Takahashi (1993). The number of the smallest fragments (25 μm) seems to be the only parameter which depends
on the CKE. A parametrization of the size of the fragments produced by graupel-graupel with dendrites collisions is therefore performed. Furthermore, bare graupel-graupel produced only less than 20 fragments per collision. This observation highlights the lower efficacy of rimed surfaces compared to vapor-grown crystals surfaces in generating fragments by collisions.

Several non-realistic aspects of Takahashi et al. (1995) experiments such as the size of the ice particles were pointed out in Korolev and Leisner (2020). These aspects have been considered in designing our experiments by generating more realistic
sizes of graupel particles, and by using an updraft during the riming and dendritic growth processes.

The dendritic crystals grown on the surface of graupel enables the production of many fragments during collisions, differing from a completely rimed surface. Future studies are required to investigate how this transition (observed in  Korolev et al., 2004) can affect collision fragmentation at different humidity and temperature conditions.

In the second series of experiments the collisions of a 4 mm graupel and a dendritic ice crystal aggregate of 10 mm diameter
as proxy for a snowflake were studied. The snowflake was manually produced by sticking dendritic ice crystal monomers together. This method can be improved in the future to have more realistic particles. The graupel-snowflake collisions were investigated as both particles were freely falling inside a fall tube. All fragments resulting from the collisions were recorded by a holographic imaging instrument. The results show that the number of fragments produced is between 50 and 450, depending on the CKE and the position where the graupel hit the snowflake. In accordance to the fragment size distribution of Vardiman
(1978), we found one mode of the FSD at 400 μm but also a second one at 50 μm for half of collisions. New fit coefficients for the parameterization of Phillips et al. (2017) were derived and presented in Table 3. The new fits revealed a very good agreement of the theoretical formulation with our experimental data both for graupel-graupel and graupel-snowflake collisions. Comparing the parameters for graupel-graupel with dendrites collisions to those for graupel-snowflake collisions, it becomes

apparent that the fragility asperity coefficient $C$ and the shape parameter $\gamma$ are similar in the two different series of experiments. However, the number of asperities per unit area $A(M)$ was lower for graupel-snowflake collisions. We note also, that the revised coefficients result in a higher number of fragments when compared to the coefficients used in the original Phillips et al. (2017) parameterization.

| Collision type | A(M) (m$^{-2}$) | C (J$^{-1}$) | $\gamma$ |
|---|---|---|---|
| Graupel-graupel with dendrites | $1.9 \times 10^8$ | $1.0 \times 10^8$ | 0.78 |
| Bare graupel-graupel | $6.4 \times 10^6$ | $9.7 \times 10^5$ | 0.55 |
| Graupel-snowflake | $2.0 \times 10^7$ | $5.8 \times 10^8$ | 0.78 |

**Table 3.** Re-fitted parameters of Phillips et al. (2017) theoretical formulation for ice-ice collisional fragmentation (see Eq. 2) based on our experiments for graupel-snowflake, graupel-graupel with dendrites, and bare graupel-graupel collisions.

Nevertheless, it is important to note that the present conditions, characterized by high ice supersaturation and large particle size, may not be representative for most ice crystals in clouds. To overcome this limitation, it is necessary to conduct future experiments with technical improvements to explore collisions at lower ice supersaturation levels and with smaller aggregate sizes. We presume that our results are more representative for fragmentation occurring above water saturation, where fragile ice crystals tend to form. To apply our results to a microphysics scheme, it is crucial to consider these factors for precautionary purposes.

As only a few studies has been carried out on the fragmentation of ice crystals by collisions, this study is a step towards a better understanding of the fragmentation breakup of ice crystals in collisions where at least one particle have dendritic (i.e. fragile) ice crystal on its surface. Our results highlight the necessity for further investigation of collision induced fragment production of atmospheric ice particles, and the dependence of the process on temperature, rimed fraction, particle size and type, and dendritic growth time. However, achieving collisions of a full range of ice crystal types under a wide range of atmospheric conditions in the laboratory remains a challenge. Nonetheless, such a realization would be beneficial to models that represent this variety of conditions. By means of improved experimental designs it might even become possible to use natural snowflakes after capturing them in the open atmosphere.

*Data availability.* The data set used for generating the figures is available under https://doi.org/10.5281/zenodo.7877368. The raw measurement data will be provided upon request.

*Video supplement.* A video supplement showing the records of the graupel-graupel, graupel-snowflake (edge), and graupel-snpwflake (central) collisions can be downloaded from https://doi.org/10.5446/62064, https://doi.org/10.5446/62065, and https://doi.org/10.5446/62066, respectively.

## Appendix A:  Graupel free fall equations

The equation of motion for a spherical liquid drop from Pruppacher and Klett (2010) can be used to describe the motion of a lump graupel which is almost spherical with

$$m_g \frac{dv}{dt} = (m_g + m_a)g - \frac{1}{2}\rho_a C_d A v^2 = \left(1 - \frac{\rho_a}{\rho_g}\right)g - \frac{3}{8}\frac{C_d \rho_a}{\rho_g r}v^2. \tag{A1}$$

where $m_g$ and $m_a$ the graupel and air masses, $\rho_g$ and $\rho_a$ are the graupel and air densities, $g$ the acceleration, $A$ the graupel cross section, $C_d$ the drag coefficient, $r$ the radius of the graupel and $v$ the graupel fall speed. The previous equation can be simplified with $C_1$ and $C_2$ terms:

$$\frac{dv}{dt} = C_1 - C_2 v^2. \tag{A2}$$

Equation A2 have to be integrated to deduce the speed and the position of the graupel during the fall speed as

$$t = \int \frac{1}{C_1 - C_2 v^2}dv = \int \frac{1}{(\sqrt{C_1} - \sqrt{C_2}v)(\sqrt{C_1} + \sqrt{C_2}v)}dv \tag{A3}$$

$$t = \int \frac{1}{2\sqrt{C_1}(\sqrt{C_1} + \sqrt{C_2}v)} + \frac{1}{2\sqrt{C_1}(\sqrt{C_1} - \sqrt{C_2}v)}dv \tag{A4}$$

$$t = \frac{1}{2\sqrt{C_1}}\left(\frac{\ln(\sqrt{C_1} + \sqrt{C_2}v)}{\sqrt{C_2}} - \frac{\ln(\sqrt{C_1} - \sqrt{C_2}v)}{\sqrt{C_2}}\right) = \frac{1}{2\sqrt{C_1 C_2}}\ln\left(\frac{1 + \sqrt{C_2}/\sqrt{C_1}v}{1 - \sqrt{C_2}/\sqrt{C_1}v}\right). \tag{A5}$$

From Eq. A5 and the $\mathrm{arctanh}$ function properties

$$t = \frac{1}{\sqrt{C_1 C_2}}\mathrm{arctanh}\left(\frac{\sqrt{C_2}}{\sqrt{C_1}}v\right). \tag{A6}$$

By integrating Eq. A6, one can deduce the speed of the graupel depending on the time

$$v = \frac{\sqrt{C_1}}{\sqrt{C_2}}\tanh\left(\sqrt{C_1}\sqrt{C_2}t\right). \tag{A7}$$

To know the position of the graupel according to the time, equation A7 have to be integrated. With the properties of the $\tanh$ function, the position of the graupel is expressed by

$$x = \frac{1}{\sqrt{C_1}}\ln\left(\cosh\left(\sqrt{C_1 C_2}t\right)\right). \tag{A8}$$

If $t \to \infty$, the fall velocity becomes equal to the terminal velocity $v_T$ of the graupel:

$$v_T = \sqrt{\frac{8r(\rho_a/\rho_g - 1)g}{3C_d}}. \tag{A9}$$

## Appendix B: Fragments size distributions

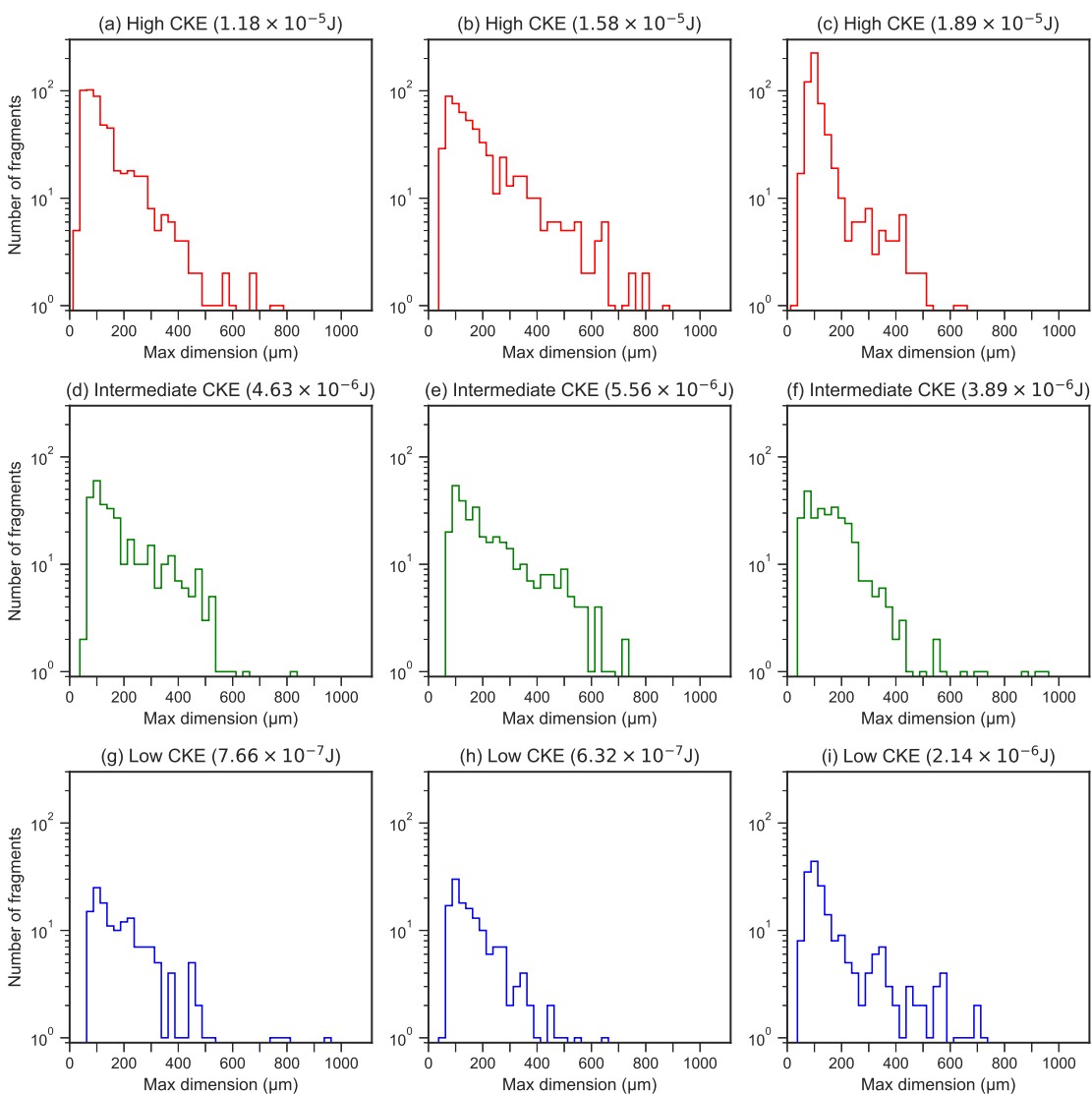

**Figure B1.** Size distributions of ice fragments produced by graupel-graupel with dendrites collisions. Low CKE in blue, intermediate CKE and high CKE in red.

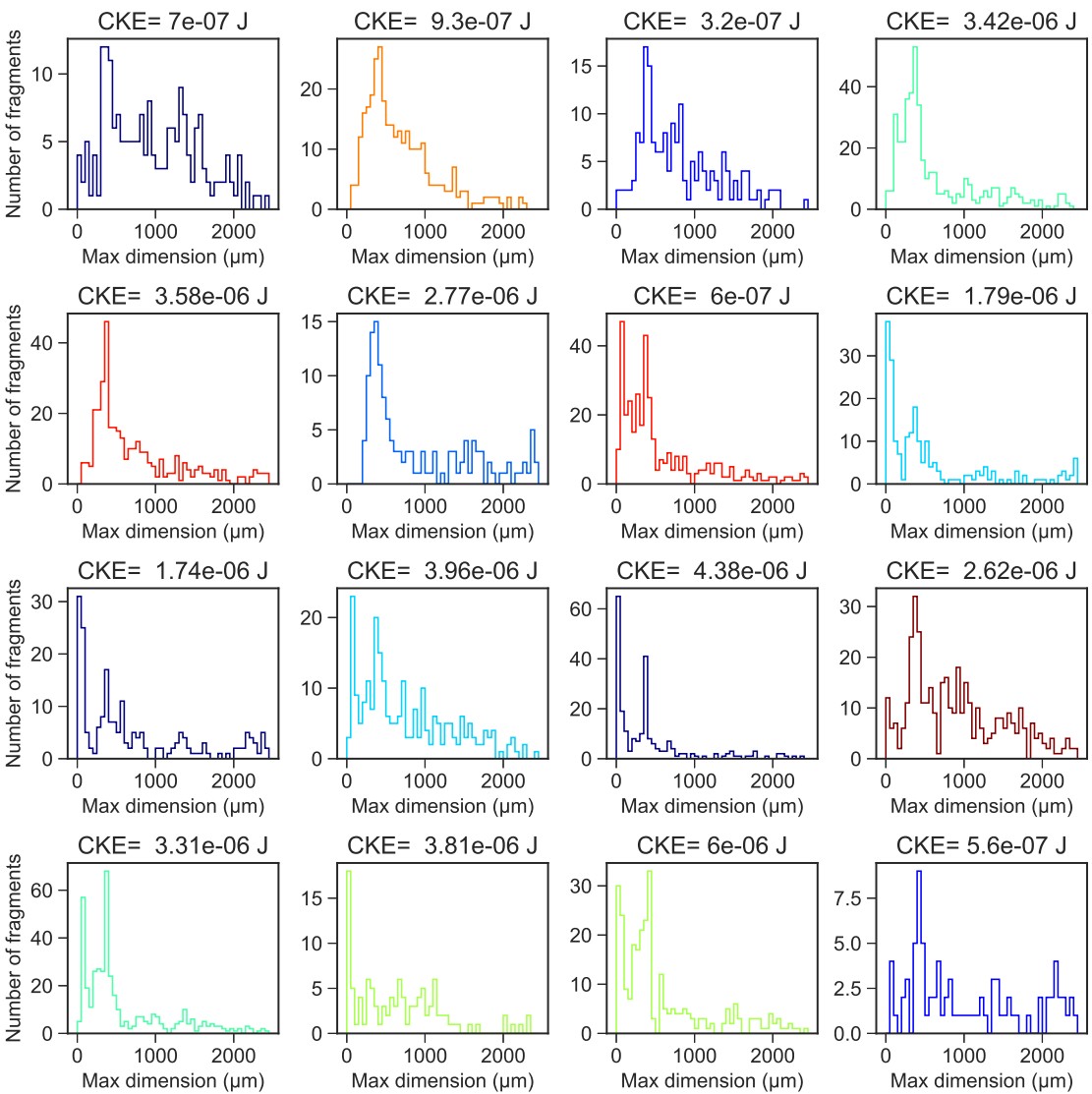

**Figure B2.** Size distributions of ice fragments produced by graupel-snowflake collisions for different impact positions (see color scale of Fig. 14).

## Appendix C:  Distributions for the cross sectional areas of the fragments

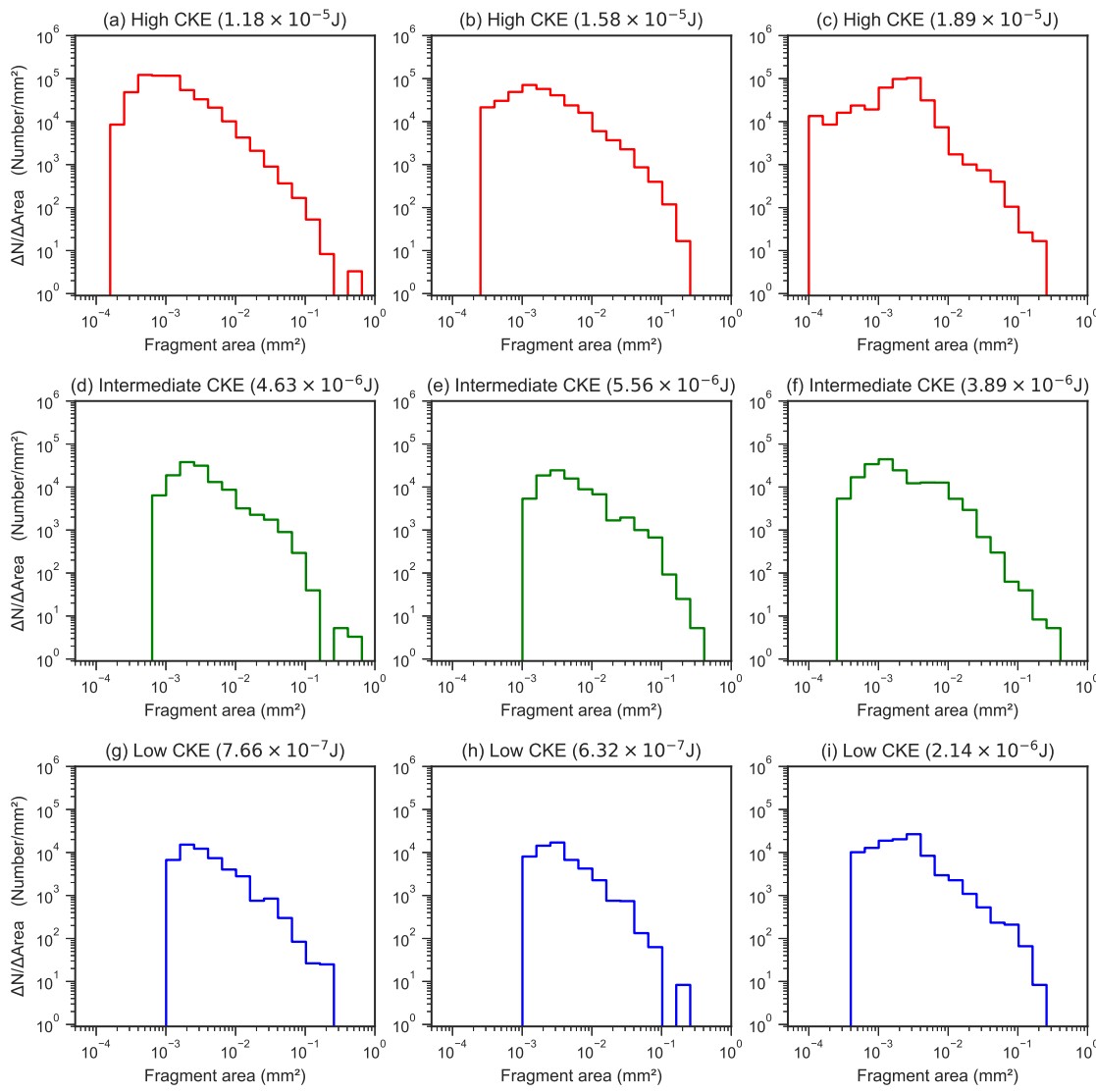

**Figure C1.** Area distributions of ice fragments produced by graupel-graupel with dendrites collisions. Low CKE in blue, intermediate CKE and high CKE in red.

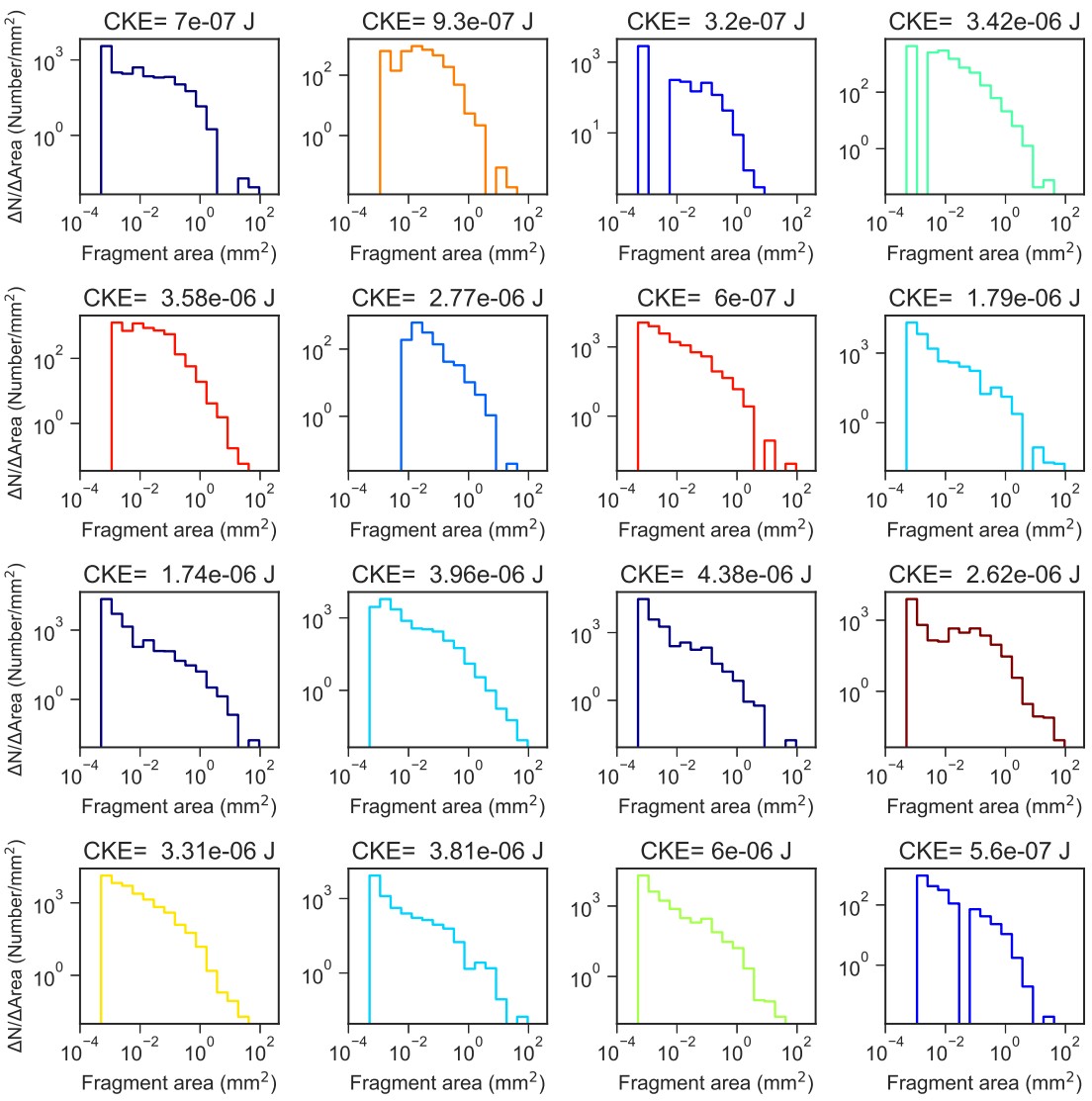

**Figure C2.** Area distributions of ice fragments produced by graupel-snowflake collisions for different impact positions (see color scale of Fig. 14).

 **Appendix D: Distributions of fragments aspect ratio**

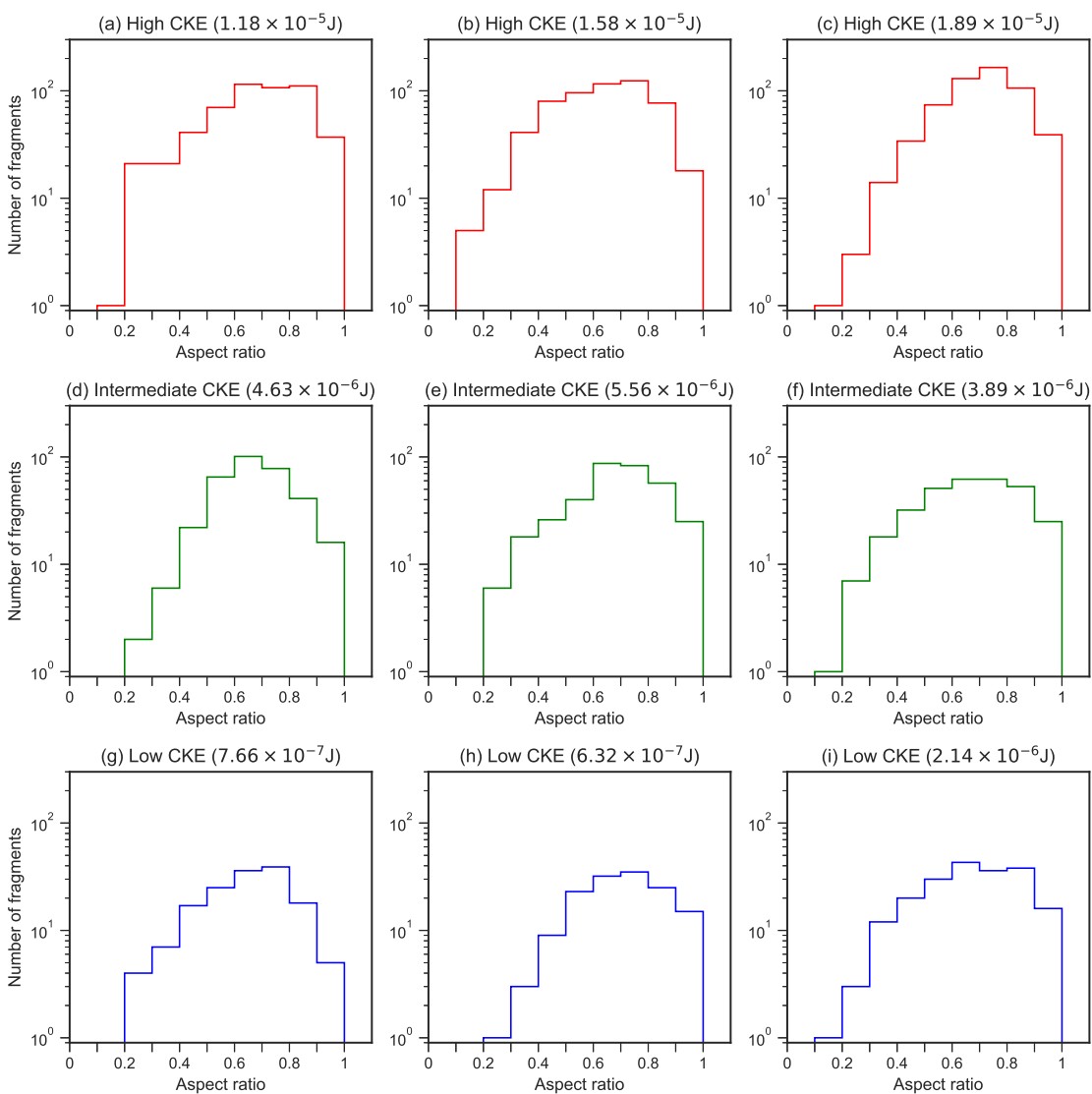

**Figure D1.** Aspect ratio (AR=$D_{min}/D_{maj}$) distributions of ice fragments produced by graupel-graupel with dendrites collisions. Low CKE in blue, intermediate CKE and high CKE in red.

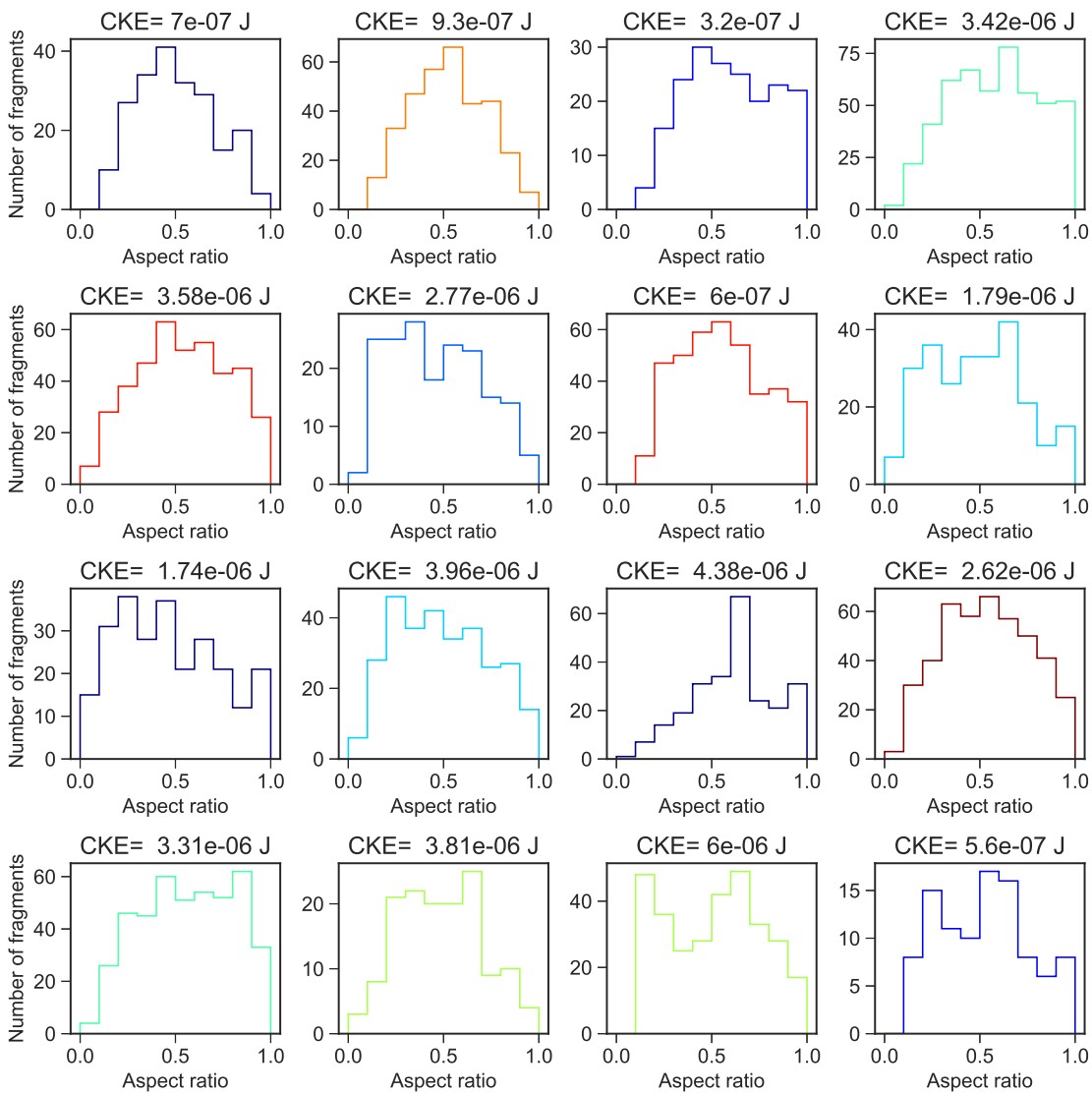

**Figure D2.** Aspect ratio (AR=$D_{min}/D_{maj}$) distributions of ice fragments produced by graupel-snowflake collisions for different impact positions (see color scale of Fig. 14).

*Author contributions.* The paper was written by P.G. and M.S. with the support and assistance of all co-authors; SB, made significant contributions by providing comments on the results, discussion, and conclusion; P.G. and F.Z. performed graupel growth experiments; P.G. carried out collision experiments, and evaluated the data; S.Y. performed ice crystal growth experiments and analyzed the data; F.Z. evaluated and analyzed the holographic data; A.T. constructed the graupel generator, designed the graupel growth and dendritic crystal growth

experiments, and holographic measurements; S.K.M. and P.G. designed the graupel growth, crystal growth and collision experiments; M.S. designed the experiments, analyzed the data.

*Competing interests.* The authors declare no competing interest.

*Acknowledgements.* We gratefully acknowledge the funding of the German Research Foundation (DFG) to initialize the special priority programme on the Fusion of Radar Polarimetry and Atmospheric Modelling (SPP-2115, PROM). The work of contributing authors was carried out in the framework of the project "FRAGILE: Exploring the role of FRAGmentation of ice particles by combining super-partIcle modelling, Laborotary studies, and polarimEtric radar observations" (Grant KN 1112/5). We also gratefully acknowledge funds from internal Max Planck Institute for Chemistry (Mainz, Germany) budget. The first author is now funded by the French National Research Agency (ANR) (ACME ANR-21-CE01-0003 project contract). We thank the reviewers of the manuscript for their invaluable comments on the constraints of our experiments. Figures 1, 2, 6, and 9a have been created with BioRender.com.

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
