# Peer review of "Fragmentation of ice particles: laboratory experiments on graupel-graupel and graupel-snowflake collisions"

_EGUsphere, 2023_

## Referee Comment (RC2)

**Review of "Fragmentation of ice particles: laboratory experiments on graupel-graupel and graupel-snowflake collisions" by Grzegorczyk and colleagues**

**Verdict**

The paper should be published only after major modifications have been made.

**Major Comments**

The present graupel and snow particles are artificial and limitations of representativeness of the lab data arise. The fact that the snowflakes were created by manually clumping together some dendritic crystals needs to be mentioned in the concluding section and in the abstract.

Although high numbers of fragments are reported for graupel-graupel and graupel-snow collisions, the morphology of the artificial particles observed are extreme. There is a lack of representativeness. For example, the snow particles studied are 1 cm wide. But most snow particles are smaller than this in any size distribution. Also, the fragility of the snow particle and the number of monomers near the collision path of the incident graupel will increase with the snowflake size.

In reality, the proposed parameterization (Eq 3) does not necessarily apply to most snow particles, because a crucial quantity is missing: area of contact. These limitations need to be discussed in the concluding section.

The proposed parameterization should be adapted to apply to a wider range of particle sizes if possible. Area of contact could be introduced as a multiplying factor into Eq (3).

I think the title should be changed to convey the fact that the particles being studied are artificial and this should also be highlighted in the abstract. The abstract and conclusions sections need to state clearly the sizes of particles studied.

It needs to be specified under what conditions of LWC and temperature the vapour growth can prevail such that the dendritic crystal can grow on the graupel, so that the graupel-graupel results are valid.

**Detailed Comments**

Line 36: Other modeling studies can also be cited that use this breakup scheme: Waman et al. (2022, JAS), Sotiropolou et al. (2021, 2022), Zhao et al..

Line 49: It is not true that Phillips et al. wrote that use of a fixed target could falsify results. In fact, they argued the opposite:

*"On the one hand, for head-on collisions the fixing of the target boosted the initial CKE without appreciably altering the energy-based coefficient of restitution q governing fragmentation. In the present paper, the laboratory observations were used only by relating fragment numbers to the initial CKE, so there is no problem in this respect.".*

It is important to read the papers that are cited.

Line 76-77:  It is not true that *both* colliding spheres were fixed during and after collision.  Phillips et al. never wrote that.  Only one of the colliding spheres was fixed.  Of course, this artificially boosted the CKE.  But as noted above, that is not really a problem, if the analysis is done in terms of CKE, relating it to the number of fragments.

Line 282:  This Equation (3) is simplistic because it neglects the role of the area of contact during impact, which depends on the particle sizes.

Line 283-284:  The maximum emission of fragments beyond a certain CKE was not merely "expected", but rather was observed in Takahashi's published data when analysed by Phillips et al. (2017) in terms of CKE.

Line 304:  What is really needed for use of the graupel-graupel results is the critical LWC and temperature range, for which the dendritic growth prevails at the surface.  Outside of these conditions, there will be no fragmentation because the surface will be rimed and any depositionally grown ice will be continually buried by fresh rime.

Line 386:  There was no intention to "rime" (accretion of supercooled droplets) the ice spheres in the Takahashi et al. lab experiment.   The purpose of their controlled supply of supercooled cloud-liquid was to control the time of exposure to high humidities and vapour growth of ice.

Line 375-400:  The concluding section needs to discuss the limitations arising from the fact that all particles studied in the present paper are artificial.  What conditions of LWC and duration of exposure are needed for graupel in a simulation to be representative of the artificial graupel observed here ?  The artificial manner of creation of these particles must be discussed.

---

## Referee Comment (RC3)

**Review of "Fragmentation of ice particles: laboratory experiments on graupel-graupel and graupel-snowflake collisions" by Grzegorczyk,et al.**

**Comments**

Secondary ice production continues to be one of the most controversial cloud microphysical processes generating a wide variety of speculations and dubious explanations. In many ways, this lack of clarity is a function of the significant challenges that exist in obtaining direct in-situ observations and running laboratory studies of secondary ice production processes. The existing laboratory studies of SIP mechanisms are sparse, and many have presented controversial results. Therefore, any attempt made in exploring the efficiency of SIP mechanisms under controlled environmental conditions is greatly appreciated. And this is particularly true for the present work, aimed at the laboratory study of the ice-ice collisional break-up SIP mechanism with a focus on graupel-graupel and graupel-snowflake collision. The laboratory setup sounds reasonable, and it can be used as a basis for subsequent studies of the efficiency of the ice-ice collisional break-up SIP mechanism for different environmental conditions and ice particle shapes. The paper is well-written and should be accepted for publication in ACP. However, I have two comments which should be included in the paper in the form of disclaimers prior to publication.

1. I have a serious concern regarding the parameterization of the ice-ice collisional breakup SIP solely based on CKE. Besides the CKE, the number of fragments generated after collision depends on the mechanical properties of the colliding particles. The mechanical properties of ice particles depend on the history of the environmental condition that this particle experienced in the past. Thus, for the sake of argument, assuming that the mass of the four graupel particles in the picture below is the same, their collision with other graupel will result in a different number of fragments, even though the CKE will be the same.

[Figure]

[Figure]

[Figure]

[Figure]

   The morphology and mechanical properties of the graupel surface depend on many parameters such as DSD, LWC, T, P, vertical wind, the graupel's mass, and density. Within the same cloud, graupel may experience a variety of time histories of the above mentioned parameters, which can subsequently generate an infinite number of possible combinations of collisional events between graupel with different mechanical properties of surface ice but having the same CKE.

   In the frame of the present study, the fragment size distributions (FSD) and their dependence on CKE (Figs.11-13) were obtained for the graupel formed under approximately the same environmental conditions as described in section 2. Therefore,

the obtained parameterization (Eq.3) describes secondary ice production for the specific graupel generated in this lab setup, and it cannot be expanded to the entire variety of possible graupel-graupel collisions. This limitation of the obtained parameterization should be clearly stated in the paper in order to mitigate the use of the obtained SIP parameterization in cloud simulations.

2. The relevance of the environmental conditions employed in the laboratory setup during the depositional growth of ice is another point of concern in this study. As described in section 2.2, that at the location of graupel, the supersaturation over ice and temperature varied in the ranges $20\% < S_i < 27\%$ and $-15C < T < -13C$, respectively. Such supersaturation over ice corresponds to up to 10% supersaturation of liquid. This is an overly high supersaturation, which normally does not occur in natural clouds, with the exception of short periods of time in vigorous updrafts. The mechanical properties of ice grown at high supersaturation are expected to be different as compared to growth at low supersaturation (e.g., below water saturation) due to an increased number of dislocations (hopper ice growth). The depositional growth of the graupel surface at lower and more realistic supersaturation is slower and may not develop protruding ice shapes (e.g., [https://doi.org/10.1175/1520-0450(2004)043%3C0612:LAISOO%3E2.0.CO;2)](https://doi.org/10.1175/1520-0450(2004)043%3C0612:LAISOO%3E2.0.CO;2), which is expected to affect the FSD and SIP efficiency. The effect of high supersaturation and relevancy of the environmental condition should be discussed in the paper as well.

**Minor comment:**  Line 102:  Rb4 => R4b

**Recommendation**: I recommend the paper for publication after adding disclaimers as discussed above.

Alexei Korolev

---

## Author Comment (AC1)

We would like to thank all reviewers for the useful comments and suggestions, which definitely helped us to improve the manuscript. Hereby we provide a detailed response to the comments and questions raised by Reviewer #1.

*Reviewer's comment:*
**I would suggest that highly dendritic growth on a graupel particle seen in fig 5b would not be observed in the real atmosphere. It would be difficult to obtain saturations above water saturation (~14% at -14C). At water saturation there would be droplets that would continue the riming.**
**For a realistic graupel with vapour grown surface ice I think that a superaturation with respect to ice but below water saturation is required.**

*Authors' response:*
It is true that the saturations were high in our experiments which might result in more fragile dendritic structures due to fast growing of ice crystals. Thus, the number of fragments might be lower at lower supersaturations, i.e. under more common atmospheric conditions. We add the following text to line 321 "*The growth of dendrites on the graupel surface that occurs under high supersaturation conditions is faster than at low supersaturation, and therefore, may result in a more fragile ice crystal structure. This might lead to more fragments produced by graupel-graupel with dendrites collisions compared to ice crystals growing at lower humidity. Cloud graupel may experience several growth processes that influence their surface properties, making their fragility dependent on their growth history. Consequently, graupel collisions of the same size, with the same collision kinetic energy, can yield different fragment numbers due to their distinct surface properties*".

From Korolev et al, 2004 it seems to be apparent that frozen drops remain spherical during vapor deposition growth if the supersaturation is less than half of that of water. Hence, we can suppose that fragmentation by collision is less effective for ice crystals staying at low humidities and more efficient for ice crystals at higher humidities where the vapor growth enables the production of long branches. Therefore, we added a sentence to line 467 "*The dendritic crystals grown on the surface of graupel enables the production of many fragments during collisions, differing from a completely rimed surface. Future studies are required to investigate how this transition (observed in Korolev et al, 2004) can affect collision fragmentation under different humidity and temperature conditions. *"

*Reviewer's comment:*

**It would be good to caveat the results for the dendrite-covered graupel vs graupel collisions.**

*Authors' response:*

We added the following text to line 479 to caveat our results "*Nevertheless, it is important to note that the present conditions, characterized by high ice supersaturation and large particle size, may not be representative for most ice crystals in clouds. To overcome this limitation, it is necessary to conduct future experiments with technical improvements to explore collisions at lower ice supersaturation levels and with smaller aggregate sizes. We presume that our results are more representative for fragmentation occurring above water saturation, where fragile ice crystals tend to form. To apply our results to a microphysics scheme, it is crucial to consider these factors for precautionary purposes.*"

*Reviewer's comment:*

**Perhaps it would be possible to include the bare graupel-graupel collision results that were alluded to?**

*Authors' response:*

We included the number of fragments generated by bare graupel-graupel collisions in Fig. 11b. We also added comments related to these results in the same section.

*Reviewer's comment:*

**line 215: What was the 'glue' used for sticking? Were the crystals just brought together at ice saturation or slightly above? How long were the crystals allowed to sinter for? I imagine there will be sensitivity to this. In the results the production process for the aggregates is mentioned, but perhaps it is worth saying that this is something that could be explored more systematically in the future?**

*Authors' response:*

The dendrites were glued by interlocking the branches of the crystals together like a natural aggregate of a cloud. This aggregation took place above the aquarium at a humidity close to that where the crystals therefore grew in an environment oversaturated with respect to the ice. We estimated that 20-60 crystals were used to form an aggregate.

We added '(*i.e. by interlocking the branches of the dendrites)*' to line 231.

It would be interesting to create a setup for the creation of aggregates in an automated way and by controlling various parameters, but this seems difficult to do. In future experiments, smaller aggregate sizes should be used to be closer to those encountered in the clouds as we mention in the revised manuscript at line 480: "*To overcome this limitation, it is necessary to conduct future experiments with technical improvements to explore collisions at lower ice supersaturation levels and with smaller aggregate sizes.*"

*Reviewer's comment:*

**line 302: Fig 12 I think needs a caveat to mention that these results are likely to be an upper bound because of the very high saturations the graupel were exposed to. Much more than is likely in a real cloud.**

*Authors' response:*
In figure caption we added *'at high supersaturation'* for the caption of Fig. 11 and 12. Furthermore, we added the following text at line 321 "*The growth of dendrites on the graupel surface that occurs under high supersaturation conditions is faster than at low supersaturation, and therefore, may result in a more fragile ice crystal structure. This might lead to more fragments produced by graupel-graupel with dendrites collisions compared to ice crystals growing at lower humidity. Cloud graupel may experience several growth processes that influence their surface properties, making their fragility dependent on their growth history. Consequently, graupel collisions of the same size, with the same collision kinetic energy, can yield different fragment numbers due to their distinct surface properties*".

*Reviewer's comment:*

**line 344: For fig. 14 graupel-snow collisions it may be appropriate to just suggest a mean and range (e.g. 200 splinters ranging from 100-400 to capture 95% of measurements. Hopefully, later experiments will provide enough data to parameterise the degree of separation effect. For models some average would likely be necessary to use.**

*Authors' response:*

This is planned to be done in a future study where more collisions can be made for statistical significance. We also provide a fit of our experiments using Eq. 2 which can be taken as a means from all experiments. More data are needed to clearly parametrize the effect of edge/central collisions.

*Reviewer's comment:*

**line 350: The size distribution of fragments is very welcome. For implementation in a model this would likely need to be scaled by the size of the snowparticle being collided with. In the first instance the use of 10mm snow aggregates could be used to scale the x axis (at least as an extra axis)?**

*Authors' response:*

Yes, thank you for this suggestion! We added the following sentence to line 350: "*However, in a first instance, we suggest that the parameters μ and σ of the FSD can be interpolated or rescaled considering the size of the parent particle (2 mm graupel here and 10 mm snowflake in section 4.2)*"

*Reviewer's comment:*

**line 354: Only 16 distributions in here so difficult to draw too many conclusions from the individual modes - apart from the mode at 50um being the size of the monomers.**

*Authors' response:*

We agree. Therefore we added to line 391: '*However, only 16 FSD are presented here, more experiments have to be done to clarify the observation mentioned before*. '

*Reviewer's comment:*

**line 368: This 50um mode is just the size of the monomers used to construct the aggregate, so unless Vardiman constructed their snow in the same way there is unlikely to have been a mode in those observations?**

*Authors' response:*

We think that the monomers forming the initial snowflake are larger, they are rather millimeter-size crystals. The 50 μm fragments have no real structures, they are just irregular ice crystal fragments. Thus, 50 μm are probably coming from something else than the monomers. Vardiman used natural ice particles to do his experiments and used an old camera which probably didn't detected small fragments (e.g 50 μm diameter ice crystals) if they were present.

We added to line 402: 'This can be due to the detection method used in Vardiman (1978) which was probably not able to detect such small fragments, or due to the different particle and experimental setup used for collisions.'

*Reviewer's comment:*

**line 401: Agreed. It would be great to see results for a range of graupel sizes and snow sizes to cover the phase space required in a numerical cloud model.**

*Authors' response:*

Thank you for this motivating comment. We are planning to extend the experiment to other graupel and snow sizes. For that, it is necessary to improve the routine of our experiments to conduct a large number of collisions.

---

## Author Comment (AC2)

We would like to thank all reviewers for the useful comments and suggestions, which definitely helped us to improve the quality of our manuscript. Hereby we provide a detailed response to the comments and questions raised by Reviewer #2. (The original comments of the reviewer are written using bold fonts, while our responses are with normal fonts.)

*Replies to the major comments of the reviewer*

**The present graupel and snow particles are artificial and limitations of representativeness of the lab data arise. The fact that the snowflakes were created by manually clumping together some dendritic crystals needs to be mentioned in the concluding section and in the abstract.**

Following the Reviewer's suggestion, we added in the abstract (line 6):

*"The particles were synthetically generated within a cold room through two distinct methods: riming and vapor deposition for graupel with diameters of 2 mm and 4 mm, and by manually sticking vapor grown ice which were generated above a warm bath to form snowflakes with a diameter of 10 mm."*

Further, we added in the conclusion (line 479):

*"Nevertheless, it is important to note that the present conditions, characterized by high ice supersaturation and large particle size, may not be representative for most ice crystals in clouds. To overcome this limitation, it is necessary to conduct future experiments with technical improvements to explore collisions at lower ice supersaturation levels and with smaller aggregate sizes. We presume that our results are more representative for fragmentation occurring above water saturation, where fragile ice crystals tend to form. To apply our results to a microphysics scheme, it is crucial to consider these factors for precautionary purposes."*

And at line 471 we added :

*"The snowflake was manually created by sticking dendritic ice crystals monomers together, this method can be improved in the future to have more realistic particles."*

**Although high numbers of fragments are reported for graupel-graupel and graupel-snow collisions, the morphology of the artificial particles observed are extreme. There is a lack of representativeness. For example, the snow particles studied are 1 cm wide. But most snow particles are smaller than this in any size distribution. Also, the fragility of the snow particle and the number of monomers near the collision path of the incident graupel will increase with the snowflake size.**

Of course, laboratory experiments cannot completely represent collisions of particles as occur in clouds. Nevertheless, one important aspect for us was to carry out graupel-graupel and graupel - snowflake collisions in free fall. This limited us to using large particles for the moment for technical reasons. We intend to conduct collision experiments with particles of smaller sizes in the future. The limitations of our current experiments are mentioned in the revised manuscript in line 321: "*The growth of dendrites on the graupel surface that occurs under high supersaturation conditions is faster than at low*

*supersaturation, and therefore, may result in a more fragile ice crystal structure. This might lead to more fragments produced by graupel-graupel with dendrites collisions compared to ice crystals growing at lower humidity. Cloud graupel may experience several growth processes that influence their surface properties, making their fragility dependent on their growth history. Consequently, graupel collisions of the same size, with the same collision kinetic energy, can yield different fragment numbers due to their distinct surface properties."*

The fact that the snowflake is large (1 cm) can be at least partly fixed using Phillips et al (2017) theoretical formulation. Nevertheless, we addressed this constraint in the new section "Limitations of the experiments"

**In reality, the proposed parameterization (Eq 3) does not necessarily apply to most snow particles, because a crucial quantity is missing: area of contact. These limitations need to be discussed in the concluding section. The proposed parameterization should be adapted to apply to a wider range of particle sizes if possible. Area of contact could be introduced as a multiplying factor into Eq (3).**

We decided to remove Eq. 3 and to replace it by Eq. 2 which corresponds to the fit our results on the theory of Phillips et al. (2017) and considers the area of the smallest particle. This allows us to rescale our results in terms of particle size and apply our results to smaller sizes of particles.

**I think the title should be changed to convey the fact that the particles being studied are artificial and this should also be highlighted in the abstract. The abstract and conclusions sections need to state clearly the sizes of particles studied.**

The fact that the ice particles were generated artificially inside the cold room is now added in the abstract '*The particles were synthetically generated in the cold room through two distinct methods: riming for graupel with diameters of 2 mm and 4 mm, and by manually sticking vapor grown ice which were generated above a warm bath to form snowflakes with a diameter of 10 mm*.'

Furthermore, we added to Conclusion:

*"In the second series of experiments the collisions of a 4 mm graupel and a dendritic ice crystal aggregate of 10 mm diameter as proxy for a snowflake were studied. The snowflake was manually created by sticking dendritic ice crystals monomers together, this method can be improved in the future to have more realistic particles"*

The size of graupel particle for graupel collisions were already mentioned in the conclusion.

Adding the word 'artificial' or a synonymous term to the title might imply that our crystals are not composed of ice; but instead, they might originate from a different material, e.g., employing 3D printing technique. This could potentially confuse the readers. In contrast, other papers featuring lab-grown ice crystals don't include any terminology related to this characteristic in their titles (e.g., Takahashi et al., 1995). Moreover, lab studies working with natural (i.e. not artificially generated) particles emphasize that including the word "natural" in their title.

Furthermore, in our opinion, the inclusion of the term "artificial" could undermine the credibility of our work and negatively impact the interest for our article. This would be regrettable, particularly considering the scarcity of existing publications on fragmentation during ice-ice collisions. We rather

added the quoted elements of the lines to inform the readers of the particle creation process. For the reasons discussed above we decided not to modify the title of our manuscript.

**It needs to be specified under what conditions of LWC and temperature the vapour growth can prevail such that the dendritic crystal can grow on the graupel, so that the graupel-graupel results are valid.**

Thank you for this comment. We added a sentence about that in line 467: "The dendritic crystals grown on the surface of graupel enables the production of many fragments during collisions, differing from a completely rimed surface. Future studies are required to investigate how this transition (observed in Korolev et al, 2004) can affect collision fragmentation under different humidity and temperature conditions. "From Korolev et al, 2004 it seems that the drops remain spherical if the supersaturation is less than half of that of water. Hence, we can suppose that fragmentation by collision is less effective for ice crystals staying at low humidities and more efficient for ice crystals at higher humidities where the vapor growth prevails.

*Replies to the detailed comments of the reviewer*

**Line 36: Other modeling studies can also be cited that use this breakup scheme: Waman et al. (2022,JAS), Sotiropolou et al. (2021, 2022), Zhao et al..**

We extended the list of publication using this breakup parameterization scheme, like Sotiropoulou et al., 2020, 2021; Zhao et al., 2021; Huang et al., 2022; Karalis et al., 2022; Waman et al., 2022; Patade et al., 2022.

**Line 49: It is not true that Phillips et al. wrote that use of a fixed target could falsify results. In fact, they argued the opposite: "On the one hand, for head-on collisions the fixing of the target boosted the initial CKE without appreciably altering the energy-based coefficient of restitution q governing fragmentation. In the present paper, the laboratory observations were used only by relating fragment numbers to the initial CKE, so there is no problem in this respect.". It is important to read the papers that are cited.**

Sorry for the false citation and the misunderstanding statement in the original manuscript. We wanted to express that the use of a fixed target could affect the results due to the non-rotation of the particles after the collision. However, it is apparent that this aspect created a debate. We therefore modified our sentence in line 52 to " *Furthermore, Korolev and Leisner (2020) pointed out that rotational energy should be considered for collisions. This is not the case in Vardiman (1978) where a fixed target was used, which may overestimate the number of generated fragments. Nevertheless Phillips et al. (2021) argue that this final rotational energy is just a small fraction of the initial CKE and that this issue can be solved applying Phillips et al. (2017) theory.* " We hope that this will allow the reader to be informed clearly about this aspect with reference to two points of view.

**Line 76-77: It is not true that both colliding spheres were fixed during and after collision. Phillips et al. never wrote that. Only one of the colliding spheres was fixed. Of course, this artificially boosted the CKE. But as noted above, that is not really a problem, if the analysis is done in terms of CKE, relating it to the number of fragments.**

We apologize for the unprecise formulation. We removed this part from the revised manuscript and changed the paragraph (line 80) to "Since the mass of the ice spheres of 1.8 cm and their contact area in Takahashi et al. (1995) experiment exceeded by far that of a natural graupel, the CKE resulted in an unnaturally large number of ice crystal fragments as highlighted by Korolev and Leisner (2020). However, Phillips et al. (2017) argue that this issue can be fixed using their theoretical scheme for fragmentation during collisions."

**Line 282: This Equation (3) is simplistic because it neglects the role of the area of contact during impact, which depends on the particle sizes.**

We agree that this equation is too simplistic and should be replaced by a more appropriate expression. This is why we introduced the Phillips et al. (2017) (line 96) equation relating the number of ice crystals to the CKE (Eq. 2 in the revised manuscript). We fit our results in terms of fragility asperity coefficient and number of asperities per surface area.

**Line 283-284: The maximum emission of fragments beyond a certain CKE was not merely "expected", but rather was observed in Takahashi's published data when analysed by Phillips et al. (2017) in terms of CKE.**

Yes, thank you for the comment. We modified the sentence (line 300) which now reads "It is expected that a maximum of ice fragments is reached at a certain CKE regarding Takahashi et al (1995) experiments and Phillips et al. (2017) theory."

**Line 304: What is really needed for use of the graupel-graupel results is the critical LWC and temperature range, for which the dendritic growth prevails at the surface. Outside of these conditions, there will be no fragmentation because the surface will be rimed and any depositionally grown ice will be continually buried by fresh rime.**

The LWC and temperature for riming process is given in table 1: -15°C and 2.2/2.3 g.m$^{-3}$

For vapor deposition: 23% ice supersaturation and -13/-15°C.

**Line 386: There was no intention to "rime" (accretion of supercooled droplets) the ice spheres in the Takahashi et al. lab experiment. The purpose of their controlled supply of supercooled cloud-liquid was to control the time of exposure to high humidities and vapour growth of ice.**

We deleted "and riming in still air" from this sentence to avoid misleading formulation.

**Line 375-400: The concluding section needs to discuss the limitations arising from the fact that all particles studied in the present paper are artificial. What conditions of LWC and duration of exposure are needed for graupel in a simulation to be representative of the artificial graupel observed here ? The artificial manner of creation of these particles must be discussed.**

Thank you also for this suggestion. We rewrote the conclusion section and added in line 479: " *Nevertheless, it is important to note that the present conditions, characterized by high ice supersaturation and large particle size, may not be representative for most ice crystals in clouds. To overcome this limitation, it is necessary to conduct future experiments with technical improvements to explore collisions at lower ice supersaturation levels and with smaller aggregate sizes. We presume that our results are more representative for fragmentation occurring above water saturation, where fragile ice crystals tend to form. To apply our results to a microphysics scheme, it is crucial to consider these factors for precautionary purposes. .*"

Furthermore, we added a new section to the manuscript that is dedicated to discuss the constraints of our experiments results.

---

## Author Comment (AC3)

We would like to thank all reviewers for the useful comments and suggestions, which definitely helped us to improve the quality of our manuscript. Hereby we provide a detailed response to the comments and questions raised by Prof. Alexei Korolev.

*Reviewer's comment:*

**I have a serious concern regarding the parameterization of the ice-ice collisional breakup SIP solely based on CKE. Besides the CKE, the number of fragments generated after collision depends on the mechanical properties of the colliding particles. The mechanical properties of ice particles depend on the history of the environmental condition that this particle experienced in the past. Thus, for the sake of argument, assuming that the mass of the four graupel particles in the picture below is the same, their collision with other graupel will result in a different number of fragments, even though the CKE will be the same.**

**The morphology and mechanical properties of the graupel surface depend on many parameters such as DSD, LWC, T, P, vertical wind, the graupel's mass, and density. Within the same cloud, graupel may experience a variety of time histories of the above mentioned parameters, which can subsequently generate an infinite number of possible combinations of collisional events between graupel with different mechanical properties of surface ice but having the same CKE.**

*Authors' response:*
We agree with these comments. It is evident that the number of fragments generated after collision depends on the mechanical properties of the particles and their environmental history, leading to different outcomes even with the same CKE. We now added to the results part this sentence to line 321: "*The growth of dendrites on the graupel surface that occurs under high supersaturation conditions is faster than at low supersaturation, and therefore, may result in a more fragile ice crystal structure. This might lead to more fragments produced by graupel-graupel with dendrites collisions compared to ice crystals growing at lower humidity. Cloud graupel may experience several growth processes that influence their surface properties, making their fragility dependent on their growth history* "

The reason we used a parameterization based solely on CKE was to easily compare our results with those of Takahashi. Certainly, such a parameterization is very bare, and several properties of the particles have to be included. We would like to stress out, however, that in laboratory experiments it is impossible to cover the whole life-cycle of a particle in a cloud. Therefore, we try to simulate the particles in terms of size, fragility, morphology, etc. In the current experiments we only used one fixed temperature (-15 °C), RH for generating the dendrites (about 115% over ice), two graupel sizes (2 and 4 mm), and three fall heights for different CKEs. Since the reviewers highlighted several constrains of our experiments, we added a separate section after Results and Discussion, in which we list and discuss such limitations.

*Reviewer's comment:*

**In the frame of the present study, the fragment size distributions (FSD) and their dependence on CKE (Figs.11-13) were obtained for the graupel formed under approximately the same environmental conditions as described in section 2.**

**Therefore, the obtained parameterization (Eq.3) describes secondary ice production for the specific graupel generated in this lab setup, and it cannot be expanded to the entire variety of possible graupel-graupel collisions. This limitation of the obtained parameterization should be clearly stated in the paper in order to mitigate the use of the obtained SIP parameterization in cloud simulations.**

*Authors' response:*
We agree with these concerns regarding the parameterization of ice-ice collisional breakup based solely on CKE. We change the simple Eq. 3 to the Phillips et al (2017) parametrization which is used by many microphysics schemes. Even if this parameterization is used to extend our results to several sizes of colliding graupel pairs, the parameters used for this one remains specific to the conditions of our laboratory experiment. This is why we now mention in line 326 that *"Since the results and parameters from Eq. 2 are obtained under high humidity around -14 °C, caution in their use is essential as they only correspond to the specific environmental conditions of our experiments. To further explore the effect of graupel surface properties on fragmentation by collision, rescaling the results (i.e., varying parameters from Eq. 2 based on temperature, humidity, and growth history) would be interesting. However, further experiments should be performed since only Takahashi et al .(1995) studied the effect of temperature on the number of fragments produced by collisions."*

*Reviewer's comment:*

**The relevance of the environmental conditions employed in the laboratory setup during the depositional growth of ice is another point of concern in this study. As described in section 2.2, that at the location of graupel, the supersaturation over ice and temperature varied in the ranges 20%<Si<27% and -15C<T<-13C, respectively. Such supersaturation over ice corresponds to up to 10% supersaturation of liquid. This is an overly high supersaturation, which normally does not occur in natural clouds, with the exception of short periods of time in vigorous updrafts. The mechanical properties of ice grown at high supersaturation are expected to be different as compared to growth at low supersaturation (e.g., below water saturation) due to an increased number of dislocations (hopper ice growth). The depositional growth of the graupel surface at lower and more realistic supersaturation is slower and may not develop protruding ice shapes (e.g., https://doi.org/10.1175/1520-0450(2004)043%3C0612:LAISOO%3E2.0.CO;2), which is expected to affect the FSD and SIP efficiency. The effect of high supersaturation and relevancy of the environmental condition should be discussed in the paper as well.**

*Authors' response:*
The effect of our environmental conditions avec now added as mentioned in the first comment.

Furthermore, we added to line 467: *"The dendritic crystals grown on the surface of graupel enables the production of many fragments during collisions, differing from a completely rimed surface. Future studies are required to investigate how this transition (observed in Korolev et al., 2004) can affect collision fragmentation at different humidity and temperature conditions."*

We also added to line 479: *"Nevertheless, it is important to note that the present conditions, characterized by high ice supersaturation and large particle size, may not be representative for most ice crystals in clouds. To overcome this limitation, it is necessary to conduct future experiments with technical improvements to explore collisions at lower ice supersaturation levels and with smaller aggregate sizes. We presume that our results are more representative for fragmentation occurring above water saturation, where fragile ice crystals tend to form. To apply our results to a microphysics scheme, it is crucial to consider these factors for precautionary purposes."*

*Reviewer's comment:*

**Minor comment: Line 102: Rb4 => R4b**

*Authors' response:*
Thank you, corrected.

---

## Author Response (AR2)

**Authors' replies on referees' comments for the revised manuscript EGUSPHERE-2023-1074**

*Referee #3*

We thank Prof. Korolev for his review and the really helpful comments.

*Report #2*

**The responses to my comments are generally fine. However, the fact that the Phillips parameterisation is now re-fitted to match their data in Figure 11 is presented in a rather obscure way. More clarity is needed here.**

That the parameterization is re-fitted and match the data in Figs. 11 and 14, is now written in a more detailed way in lines 300 to 306, as well as from line 383 to 387. Also some text regarding the match between the experimental data and the parameterization has been added to Conclusion.

**The authors need to clearly state in the caption of Figure 11 (and any other figures) that the line plotted that fits well all the data is actually a re-fitting of the Phillips et al. formulation (their new equation 2).**

We added the corresponding text to the captions of Fig. 11 and Fig. 14.

**A table with the new coefficients for this fit is needed. The fact that the formulation fits the data well after re-fitting (Fig. 11) needs to be stated clearly in the conclusion section. Other readers will want to apply this re-fitted version of the Phillips formulation and this needs to be made as easy as possible with transparency in the concluding section.**

We added Table 3 to Conclusion that contains all the fit parameters for graupel-graupel and graupel snowflake collisions.

**The re-fitting of this formulation to the new data needs to be mentioned in the Abstract, because it is a salient feature of the paper. It enables modellers to apply the lab results shown in the paper to all collisions generally. Otherwise the lab results would simply be an empirical curiosity. Instead, at the moment, the only mention of the Phillips formulation in the concluding section seems to be a negative comment.**

We added the required text to the Abstract. We also expressed in Conclusion that the Phillips formulation describes our experimental data well.